# In-sample Actor Critic for Offline Reinforcement Learning

**Hongchang Zhang**[1*]**,Yixiu Mao**[1*]**,Boyuan Wang**[1]**,Shuncheng He**[1]**,Yi Xu**[2]**,Xiangyang Ji**[1]

[1]Tsinghua University    [2]Dalian University of Technology
{hc-zhang19,myx21,wangby22,hesc16}@mails.tsinghua.edu,
yxu@dlut.edu, xyji@tsinghua.edu

## Abstract

Offline reinforcement learning suffers from out-of-distribution issue and extrapolation error. Most methods penalize the out-of-distribution state-action pairs or regularize the trained policy towards the behavior policy but cannot guarantee to get rid of extrapolation error. We propose In-sample Actor Critic (IAC), which utilizes sampling-importance resampling to execute in-sample policy evaluation. IAC only uses the target Q-values of the actions in the dataset to evaluate the trained policy, thus avoiding extrapolation error. The proposed method performs unbiased policy evaluation and has a lower variance than importance sampling in many cases. Empirical results show that IAC obtains competitive performance compared to the state-of-the-art methods on Gym-MuJoCo locomotion domains and much more challenging AntMaze domains.

## 1 Introduction

Reinforcement learning (RL) aims to solve sequential decision problems and has received extensive attention in recent years (Mnih et al., 2015). However, the practical applications of RL meet several challenges, such as risky attempts during exploration and time-consuming data collecting phase. Offline RL is capable of tackling these issues without interaction with the environment. It can get rid of unsafe exploration and could tap into existing large-scale datasets (Gulcehre et al., 2020).

However, offline RL suffers from out-of-distribution (OOD) issue and extrapolation error (Fujimoto et al., 2019). Numerous works have been proposed to overcome these issues. One branch of popular methods penalizes the OOD state-action pairs or regularizes the trained policy towards the behavior policy (Fujimoto & Gu, 2021; Kumar et al., 2020). These methods have to control the degree of regularization to balance pessimism and generalization, and thus are sensitive to the regularization level (Fujimoto & Gu, 2021). In addition, OOD constraints cannot guarantee to avoid extrapolation error (Kostrikov et al., 2022). Another branch chooses to eliminate extrapolation error completely (Brandfonbrener et al., 2021; Kostrikov et al., 2022). These methods conduct in-sample learning by only querying the Q-values of the actions in the dataset when formulating the Bellman target. However, OneStep RL (Brandfonbrener et al., 2021) estimates the behavior policy's Q-value according to SARSA (Sutton & Barto, 2018) and only improves the policy a step based on the Q-value function, which has a limited potential to discover the optimal policy hidden in the dataset. IQL (Kostrikov et al., 2022) relies on expectile regression to perform implicit value iteration. It can be regarded as in-support Q-learning when the expectile approaches 1, but suffers from instability in this case. Thus a suboptimal solution is obtained by using a smaller expectile. Besides, these two lines of study adapt the trained policy to the fixed dataset's distribution.

Then one question appears-"Can we introduce the concept of in-sample learning to iterative policy iteration, which is a commonly used paradigm to solve RL"? General policy iteration cannot be updated in an in-sample style, since the trained policy will inevitably produce actions that are out of the dataset (out-of-sample) and provide overestimated Q-target for policy evaluation. To enable in-sample learning, we first consider sampling the target action from the dataset and reweighting

---

[*]Equal contribution.

the temporal difference gradient via importance sampling. However, it is known that importance sampling suffers from high variance (Precup et al., 2001) and would impair the training process.

In this paper, we propose In-sample Actor Critic (IAC), which performs iterative policy iteration and simultaneously follows the principle of in-sample learning to eliminate extrapolation error. We resort to sampling-importance resampling (Rubin, 1988) to reduce variance and execute in-sample policy evaluation, which formulates the gradient as it is sampled from the trained policy. To this end, we use SumTree to sample according to the importance resampling weight. For policy improvement, we tap into advantage-weighted regression (Peng et al., 2019) to control the deviation from the behavior policy. The proposed method executes unbiased policy evaluation and has smaller variance than importance sampling in many cases. We point out that, unlike previous methods, IAC adapts the dataset's distribution to match the trained policy during learning dynamically. We test IAC on D4RL benchmark (Fu et al., 2020), including Gym-MuJoCo locomotion domains and much more challenging AntMaze domains. The empirical results show the effectiveness of IAC.

## 2 RELATED WORKS

**Offline RL**. Offline RL, previously termed batch RL (Ernst et al., 2005; Riedmiller, 2005), provides a static dataset to learn a policy. It has received attention recently due to the extensive usage of deep function approximators and the availability of large-scale datasets (Fujimoto et al., 2019; Ghasemipour et al., 2021). However, it suffers from extrapolation error due to OOD actions. Some works attempt to penalize the Q-values of OOD actions (Kumar et al., 2020; An et al., 2021). Other methods force the trained policy to be close to the behavior policy by KL divergence (Wu et al., 2019), behavior cloning (Fujimoto & Gu, 2021), or Maximum Mean Discrepancy(MMD) (Kumar et al., 2019). These methods cannot eliminate extrapolation error and require a regularization hyperparameter to control the constraint level to balance pessimism and generalization. Another branch chooses to only refer to the Q-values of in-sample actions when formulating the Bellman target without querying the values of actions not contained in the dataset (Brandfonbrener et al., 2021; Kostrikov et al., 2022). By doing so, they can avoid extrapolation error. OneStep RL (Brandfonbrener et al., 2021) evaluates the behavior policy's Q-value function and only conducts one-step of policy improvement without off-policy evaluation. However, it performs worse than the multi-step counterparts when a large dataset with good coverage is provided. IQL (Kostrikov et al., 2022) draws on expectile regression to approximate an upper expectile of the value distribution, and executes multi-step dynamic programming update. When the expectile approaches 1, it resembles in-support Q-learning in theory but suffers from instability in practice. Thus a suboptimal solution is obtained by using a smaller expectile. Our proposed method opens up a venue for in-sample iterative policy iteration. It prevents querying unseen actions and is unbiased for policy evaluation. In practice, it modifies the sampling distribution to allow better computational efficiency. OptiDICE (Lee et al., 2021) also does not refer to out-of-sample samples. However, it involves with complex minmax optimization and requires a normalization constraint to stabilize the learning process.

**Importance sampling.** Importance sampling's application in RL has a long history (Precup, 2000) for its unbiasedness and consistency (Kahn & Marshall, 1953). Importance sampling suffers from high variance, especially for long horizon tasks and high dimensional spaces (Levine et al., 2020). Weighted importance sampling (Mahmood et al., 2014; Munos et al., 2016) and truncated importance sampling (Espeholt et al., 2018) have been developed to reduce variance. Recently, marginalized importance sampling has been proposed to mitigate the high variance of the multiplication of importance ratios for off-policy evaluation (Nachum et al., 2019; Liu et al., 2018). Sampling-importance resampling is an alternative strategy that samples the data from the dataset according to the importance ratio (Rubin, 1988; Smith & Gelfand, 1992; Gordon et al., 1993). It has been applied in Sequential Monte Carlo sampling (Skare et al., 2003) and off-policy evaluation (Schlegel et al., 2019). To the best of our knowledge, our work is the first to draw on sampling-importance resampling to solve the extrapolation error problem in offline RL.

## 3 PRELIMINARIES

**RL.** In RL, the environment is typically assumed to be a Markov decision process (MDP) $(\mathcal{S}, \mathcal{A}, \mathcal{R}, p, \gamma)$, with state space $\mathcal{S}$, action space $\mathcal{A}$, scalar reward function $\mathcal{R}$, transition dynam-

ics $p$, and discount factor $\gamma$ (Sutton & Barto, 2018). The agent interacts with the MDP according to a policy $\pi(a|s)$, which is a mapping from states to actions (deterministic policy) or a probability distribution over actions (stochastic policy). The goal of the agent is to obtain a policy that maximizes the expected discounted return: $\mathbb{E}_{\pi}[\sum_{t=0}^{\infty} \gamma^t r_t]$. Off-policy RL methods based on approximate dynamic programming typically utilize a state-action value function (Q-value function), which measures the expected discounted return obtained by starting from the state-action pair $(s, a)$ and then following the policy $\pi$: $Q(s, a) = \mathbb{E}_{\pi}[\sum_{t=0}^{\infty} \gamma^t r_t | s_0 = s, a_0 = a]$.

**Offline RL.** In offline RL, the agent is prohibited from interacting with the environment. Instead, it is provided with a fixed dataset collected by some unknown behavior policy $\beta$. Ordinary approximate dynamic programming methods evaluate policy $\pi$ by minimizing temporal difference error, according to the following loss

$$L_{TD}(\theta) = \mathbb{E}_{(s,a,s') \sim \mathcal{D}}[(r(s,a) + \gamma \mathbb{E}_{a' \sim \pi_{\phi}(\cdot|s')} Q_{\hat{\theta}}(s', a') - Q_{\theta}(s, a))^2], \tag{1}$$

where $\mathcal{D}$ is the dataset, $\pi_{\phi}$ is a policy parameterized by $\phi$, $Q_{\theta}(s, a)$ is a $Q$ function parameterized by $\theta$, and $Q_{\hat{\theta}}(s, a)$ is a target network whose parameters are updated via Polyak averaging.

We denote the $i^{th}$ transition $(s_i, a_i, r_i, s'_i, a'_i)$ in $\mathcal{D}$ as $x_i$, and $\mathcal{D} = \{x_1, \ldots, x_n\}$. For some transition $x = (s, a, r, s', a')$, let transition-wise TD update $\Delta(x)$ be the gradient of transition-wise TD error,

$$\Delta(x) = \nabla_{\theta} Q_{\theta}(s, a)(Q_{\theta}(s, a) - r(s, a) - \gamma Q_{\hat{\theta}}(s', a')).$$

For the convenience of subsequent theoretical analysis, we also define the expected value of the TD update based on the gradient of Eqn. (1) by replacing the empirical distribution of the dataset with $\beta$ induced distribution [1]

$$\Delta_{TD} = \mathbb{E}_{x \sim p_{\pi}}[\Delta(x)], \tag{2}$$

where $p_{\pi} = d^{\beta}(s, a) P(s'|s, a) \pi(a'|s')$ and $d^{\beta}(s, a)$ is the normalized and discounted state-action occupancy measure of the policy $\beta$. That is, $d^{\beta}(s, a) = (1 - \gamma) \mathbb{E}\left[\sum_{t=0}^{\infty} \gamma^t \mathbb{I}(s_t = s, a_t = a) \mid a_t \sim \pi(\cdot \mid s_t)\right]$.

Besides policy evaluation, a typical policy iteration also includes policy improvement. In continuous action space, a stochastic policy can be updated by reparameterization:

$$\phi \leftarrow \operatorname{argmax}_{\phi} \mathbb{E}_{s \sim \mathcal{D}, \epsilon \sim \mathcal{N}}\left[Q_{\theta}\left(s, f_{\phi}\left(\epsilon; s\right)\right)\right], \tag{3}$$

where $\mathcal{N}$ is a Gaussian distribution. In offline RL, OOD actions $a'$ can produce erroneous values for $Q_{\hat{\theta}}(s', a')$ in Q-value evaluation and lead to an inaccurate estimation of Q-value. Then in policy improvement stage, where the policy is optimized to maximize the estimated $Q_{\theta}$, the policy will prefer OOD actions whose values have been overestimated, resulting in poor performance. Most current methods either directly constrain policy $\pi$ or regularize OOD actions' values to constrain policy indirectly.

## 4 IN-SAMPLE ACTOR CRITIC

In the following, we introduce sampling-importance resampling and tap into it for offline RL to allow in-sample learning. Then we show that the proposed method is unbiased and has a smaller variance than importance sampling in many cases. Last, we present a practical implementation of the algorithm.

### 4.1 SAMPLING-IMPORTANCE RESAMPLING

Consider a statistical problem relevant to offline RL. Suppose we want to obtain samples from a target distribution $p(x)$, but we only have access to samples from some proposal distribution $q(x)$. How can we simulate random draws from $p$? Assuming $\text{supp}(q) \supseteq \text{supp}(p)$, a classic algorithm sampling-importance resampling (SIR) (Rubin, 1988) addresses this with the following procedure:

Step 1.(Sampling) Draw independent random samples $\{x_1, \ldots, x_n\}$ from $q$.

Step 2.(Importance) Calculate the importance ratio for each $x_i$: $w(x_i) = p(x_i)/q(x_i)$

---

[1] We omit the learning rate for simplicity

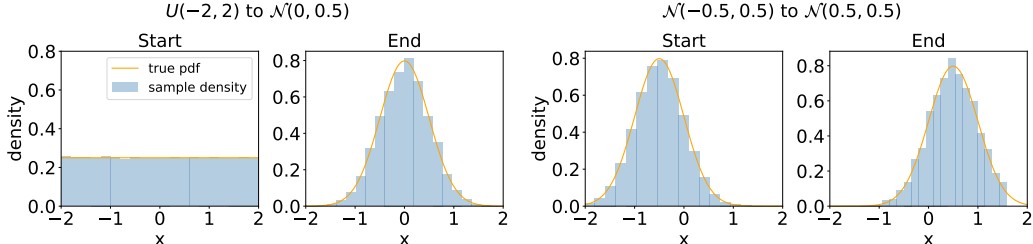

Figure 1: SIR for distribution correcting. We generate 100,000 random values from proposal distribution $q$, and use SIR to resample 10,000 random items out of it to approximate target distribution $p$. Sample histograms and the actual underlying density of both $q$(start) and $p$(end) are presented. **Left:** Uniform $U(-2, 2)$ to Gaussian $\mathcal{N}(0, 0.5)$. **Right:** Gaussian $\mathcal{N}(-0.5, 0.5)$ to another Gaussian $\mathcal{N}(0.5, 0.5)$.

Step 3.(Resampling) Draw $x^*$ from the discrete distribution over $\{x_1, \ldots, x_n\}$ with sampling probabilities $\rho(x_i) = w(x_i) / \sum_{j=1}^{n} w(x_j)$.

SIR is very similar to importance sampling(IS), except that IS samples according to $q$ and multiplies the result by the importance ratio, while SIR corrects the distribution $q$ by the importance ratio to approximate $p$. The following proposition shows the consistency of SIR: as $n \to \infty$, the resampling distribution converges to the target distribution $p$.

**Proposition 1.** *If* $\text{supp}(q) \supseteq \text{supp}(p)$, *as* $n \to \infty$, *the samples from SIR will consist of independent draws from p. Namely, As* $n \to \infty$, $x^*$ *is distributed according to p.*

All proofs could be found in Appendix B.

Fig. 1 illustrates SIR for distribution correcting on simple one-dimensional cases. Note that Fig. 1 (left) simulates the case when a dataset is generated by a uniform distribution and the target distribution is Gaussian, while Fig. 1 (right) corresponds to when a dataset is generated by some Gaussian distribution, and the target distribution is Gaussian with a different mean.

### 4.2 IN-SAMPLE ACTOR CRITIC

In this work, we adopt SARSA-style in-sample learning because in-distribution constraints widely used in prior work might not be sufficient to avoid extrapolation error (Kostrikov et al., 2022).

Our method is based on policy iteration, which consists of policy evaluation (PE) and policy improvement (PI). In PE, using in-sample actions $a' \in \mathcal{D}$ rather than $a' \sim \pi(\cdot|s')$ in the TD target introduces a bias. We consider introducing an importance ratio $w(s', a') = \pi(a'|s')/\beta(a'|s')$. Under the assumption of importance sampling: $\Delta(x_i)\pi(a_i'|s_i') = 0$ whenever $\beta(a_i'|s_i') = 0$, Eqn. (2) can be rewritten as follows

$$\Delta_{TD} = \mathbb{E}_{(s,a,s') \sim \mathcal{D}} \mathbb{E}_{a' \sim \beta(\cdot|s')} [w(s', a') \nabla_\theta Q_\theta(s, a)(Q_\theta(s, a) - r(s, a) - \gamma Q_{\hat{\theta}}(s', a'))]. \quad (4)$$

Here, the assumption of IS (as well as SIR) actually coincides with the conception of support-constrained policy set (Kumar et al., 2019):

**Definition 2** (Support-constrained policy). *Assuming the data distribution is generated by a behavior policy $\beta$, the support-constrained policy class $\Pi^\beta$ is defined as*

$$\Pi^\beta = \{\pi \mid \pi(a|s) = 0 \text{ whenever } \beta(a|s) = 0\} \quad (5)$$

This means that a learned policy $\pi(a|s)$ has a positive density only where the density of the behavior policy $\beta(a|s)$ is positive, instead of the constraint on the value of the density $\pi(a|s)$ and $\beta(a|s)$ which is overly restrictive in many cases. Previous works have demonstrated the superiority of restricting the support of the learned policy (Kumar et al., 2019; Ghasemipour et al., 2021)

In practice, it is unrealistic to use the whole dataset to empirically estimate $\Delta_{TD}$ (expected value of update) every iteration, in spite of its low variance. Consequently, we consider estimating $\Delta_{TD}$ by

sampling a mini-batch of size $k$ from the dataset. Specifically, we sample $\{\check{x}_1, \ldots, \check{x}_k\}$, where $\check{x}_j$ is sampled uniformly from $\mathcal{D} = \{x_1, \ldots, x_n\}$. It leads to an IS estimator of $\Delta_{TD}$:

$$\hat{\Delta}_{IS} = \frac{1}{k} \sum_{j=1}^{k} w(s'_j, a'_j)\Delta(\check{x}_j), \quad \check{x}_j \sim \{x_1, \ldots, x_n\} \text{ uniformly} \tag{6}$$

Though IS is consistent and unbiased (Kahn & Marshall, 1953), it suffers from high or even infinite variance due to large magnitude IS ratios (Precup et al., 2001). The high variance of $\hat{\Delta}_{IS}$ could destabilize the TD update and lead to a poor solution.

In this work, we adopt SIR instead of IS to reduce the variance and stabilize training. Specifically, we remove the IS ratio and sample $\{\tilde{x}_1, \ldots, \tilde{x}_k\}$, where $\tilde{x}_j$ is sampled from $\{x_1, \ldots, x_n\}$ with probability proportional to $w(s'_j, a'_j)$, rather than uniformly like all prior offline RL works. It leads to another SIR estimator of $\Delta_{TD}$:

$$\hat{\Delta}_{SIR} = \frac{1}{k} \sum_{j=1}^{k} \Delta(\tilde{x}_j), \quad \tilde{x}_j \overset{\rho}{\sim} \{x_1, \ldots, x_n\} \text{ with probability } \rho_j = \frac{w(s'_j, a'_j)}{\sum_{i=1}^{n} w(s'_i, a'_i)} \tag{7}$$

Intuitively, in offline RL setting, this resampling strategy reshapes the data distribution of $\mathcal{D}$ to adapt to the current policy. Unfortunately, unlike the IS estimator $\hat{\Delta}_{IS}$, $\hat{\Delta}_{SIR}$ is a biased estimator of $\Delta_{TD}$. Subsequently, we show that by simply multiplying $\hat{\Delta}_{SIR}$ with the average importance ratio in the buffer $\bar{w} := \frac{1}{n} \sum_{i=1}^{n} w_i$, we get an unbiased estimate of $\Delta_{TD}$.

**Theorem 3.** *Assume that an offline dataset $\mathcal{D}$ of $n$ transitions is sampled i.i.d according to $p_\beta(x = (s, a, r, s', a')) = d_\beta(s, a)P(s'|s, a)\beta(a'|s')$, and $\pi$ is support-constrained (i.e., $\pi \in \Pi^\beta$). Then,*

$$\mathbb{E}[\bar{w}\hat{\Delta}_{SIR}] = \Delta_{TD}$$

*where $\Delta_{TD}$ is the expected update across all transitions in $\mathcal{D}$ defined in Eqn. (2); $\hat{\Delta}_{SIR}$ is the empirical update across the sampled mini-batch defined in Eqn. (7); $\bar{w} := \frac{1}{n} \sum_{i=1}^{n} w_i$ is the average importance ratio in the dataset.*

In fact, $\hat{\Delta}_{SIR}$ gives the correct direction, and we do not need to care about the actual value of the update. The reason is that, the scalar $\bar{w}$ remains the same across all mini-batches during SGD learning, so we can include $\bar{w}$ in the learning rate and just use $\hat{\Delta}_{SIR}$ as the update estimator in PE. We point out that there is no need to adjust the conventional learning rate, because for a large enough dataset, $\bar{w}$ is close to $1$.[2]

Theorem 3 guarantees that if the policy $\pi$ is constrained within the support of behavior policy $\beta$ during learning, our method yields an unbiased policy evaluation process via in-sample learning, thus avoiding extrapolation error. Conversely, if the support of the current policy deviates much from the dataset, which is common in practice when the dataset distribution is narrow, and the trained policy is randomly initialized, Theorem 3 can not provide performance guarantees. So in PI, we implicitly enforce a constraint with advantage-weighted regression (Peters & Schaal, 2007; Peng et al., 2019), controlling deviation from the behavior policy. Since in PE, we have sampled transitions $\{\tilde{x}_1, \ldots, \tilde{x}_k\}$ non-uniformly from $\mathcal{D}$, for convenience, we use the same transitions to perform PI, instead of sampling from $\mathcal{D}$ again uniformly, leading to the following loss:

$$L_\pi(\phi) = -\mathbb{E}_{(s,a)\overset{\rho}{\sim}\mathcal{D}} \left[ \exp(\beta(Q_\theta(s, a) - \mathbb{E}_{\hat{a}\sim\pi_\phi(\cdot|s)}Q_\theta(s, \hat{a})) \log \pi_\phi(a|s) \right], \tag{8}$$

where $\overset{\rho}{\sim}$ denotes sampling from discrete distribution $\rho$.

Note that even though our method is in-sample, a constraint (implicit or explicit) is necessary due to both our theoretical requirement and empirical results of previous works. Among previous in-sample approaches, IQL (Kostrikov et al., 2022) adopts the same advantage-weighted regression in the policy extraction step, while the performance of OneStep RL (Brandfonbrener et al., 2021) will have a sharp drop without constraints. One possible reason is that in-sample methods do not update out-of-sample $(s, a)$ pairs. Their Q-values are completely determined by the initialization and generalization of the neural network, which is uncontrolled. As a result, despite that in-sample methods address the Q-value extrapolation error, vanilla PI without constraints can still choose out-of-sample actions whose Q-values are very inaccurate.

---

[2]$\bar{w} \approx \mathbb{E}_{s\sim d_\beta(s), a\sim\beta(a|s)}[\frac{\pi(a|s)}{\beta(a|s)}] = \sum_{s,a} \frac{\pi(a|s)}{\beta(a|s)}\beta(a|s)d_\beta(s) = 1$

### 4.3 LOWER VARIANCE

Theorem 3 shows that the proposed method provides unbiased policy evaluation with in-sample learning. In this section, we theoretically prove that our SIR estimator (see Eqn. (7)) has a lower variance than the IS estimator in many cases and thus yields a more stable learning process.

**Proposition 4.** *Assume that the gradient is normalized, then the following result holds,*

$$\mathbb{V}[\bar{w}\hat{\Delta}_{SIR}] \leq \mathbb{V}[\hat{\Delta}_{IS}]. \tag{9}$$

Proposition 4 indicates that when the scale of the gradient for the sample does not vary a lot across the dataset, there is a high probability that SIR will have a smaller variance than IS.

**Proposition 5.** *Assume that* $\|\Delta(x)\|_2^2$ *has a positive correlation with* $\frac{1}{\beta(a'|s')}$ *for* $x \in \mathcal{D}$ *and policy* $\pi$ *is uniform, then the following holds,*

$$\mathbb{V}[\bar{w}\hat{\Delta}_{SIR}] \leq \mathbb{V}[\hat{\Delta}_{IS}]. \tag{10}$$

In general, the sample with a large behavior distribution density usually has a small-scale gradient due to training, which corresponds to the assumption in Proposition 5.

### 4.4 PRACTICAL ALGORITHM

IAC is the first practical algorithm to adapt the dataset distribution to the learned policy, and we design the algorithm to be as simple as possible to avoid some complex modules confusing our algorithm's impact on the final performance.

**Density Estimator.** IAC requires the behavior density $\beta$ to be the denominator of importance resampling weight (Eqn. (7)). We learn a parametric estimator for the behavior policy by maximum likelihood estimation. The estimated behavior policy $\beta_\omega$ is parameterized by a Gaussian distribution. The objective of $\beta_\omega$ is

$$\max_{\beta_\omega} \mathbb{E}_{s,a\sim\mathcal{D}} \log \beta_\omega(a|s), \tag{11}$$

where $\omega$ is the parameter of the estimated behavior policy.

**Policy Evaluation.** In policy evaluation phase, IAC uses non-uniformly sampled SARSA

---

**Algorithm 1** IAC

**Input:** Dataset $\mathcal{D} = \{(s, a, r, s', a')\}$
Initialize behavior policy $\beta_\omega$, policy network $\pi_\phi$, Q-network $Q_\theta$, and target Q-network $Q_{\theta'}$
**// Behavior Policy Pre-training**
**for** each gradient step **do**
    Sample minibatch $(s, a) \sim \mathcal{D}$
    Update $\omega$ according to Eqn. (11)
**end for**
**// Policy Training**
**for** each gradient step **do**
    Sample minibatch $(s, a, r, s', a')$ proportional to $\rho(s', a')$ from $\mathcal{D}$
    Update $\theta$ by applying TD update in Eqn. (7)
    Update $\phi$ by minimizing $L_\pi(\phi)$ in Eqn. (8)
    Update target network: $\theta' \leftarrow (1-\tau)\theta' + \tau\theta$
**end for**

---

(sampling proportional to $\rho(s', a')$ ) to evaluate the trained policy. We represent the policy with a Gaussian distribution for its simple form of density.

**Policy Improvement.** In policy improvement phase, Eqn. (8) requires calculating the expectation of the Q-value concerning the current policy. We find that replacing the expectation with the Q-value of the policy's mean already obtain good performance. Also, it simplifies the training process without learning a V-function.

**SumTree.** The importance resampling weight $\rho$ is determined by $\pi(a'|s')$ and $\beta_\omega(a'|s')$. While $\beta_\omega(a'|s')$ is fixed after pretraining, $\pi(a'|s')$ changes as $\pi$ updates during training. We adopt the Sumtree data structure to efficiently update $\rho$ during training and sample proportional to $\rho$. It is similar to prioritized experience replay (PER) (Schaul et al., 2015). In PER, $\rho$ is implemented as the transition-wise Bellman error, where sampling proportional to $\rho$ replays important transitions more frequently, and thus the Q-value is learned more efficiently. In our proposed algorithm, $\rho$ is implemented as the importance resampling weight. Sampling proportional to $\rho$ provides an unbiased and in-sample way to evaluate any support-constrained policy.

**Overall algorithm.** Putting everything together, we summarize our final algorithm in Algorithm 1. Our algorithm first trains the estimated behavior policy using Eqn. (11) to obtain the behavior density. Then it turns to the Actor-Critic framework for policy training.

## 5 Discussion

### 5.1 One-step and multi-step dynamic programming

The most significant advantage of one-step approaches (Brandfonbrener et al., 2021) is that value estimation is completely in-sample and thus more accurate than multi-step dynamic programming approaches, which propagate and magnify estimation errors. On the other hand, multi-step dynamic programming approaches can also propagate useful signals, which is essential for challenging tasks or low-performance datasets. IAC belongs to multi-step dynamic programming and enjoys the benefit of one-step approaches.

To show the relationship between IAC and one-step approaches, we define a general SIR simulator:

$$\hat{\Delta}_{SIR}^{\eta} = \frac{1}{k}\sum_{j=1}^{k}\Delta(\tilde{x}_j), \quad \tilde{x}_j \overset{\rho}{\sim} \{x_1, \ldots, x_n\} \text{ with probability } \rho_j = \frac{w(s'_j, a'_j)^{\eta}}{\sum_{i=1}^{n} w(s'_i, a'_i)^{\eta}}. \tag{12}$$

Note that IAC corresponds to the case when $\eta = 1$ while it reduces to OneStep RL (Brandfonbrener et al., 2021) when $\eta = 0$. We have the following result about $\hat{\Delta}_{SIR}^{\eta}$.

**Proposition 6.** *Assume that* $\forall x \in \mathcal{D}$, $\Delta(x) = \mathbf{h}$, *where* $\mathbf{h}$ *is a constant vector. Let* $\eta \in [0, 1]$, $\bar{w}_{\eta}$ *denote* $\frac{1}{n}\sum_{j=1}^{n} w(s_j, a_j)^{\eta}$. *Assume that* $\sum_{j=1}^{n} w(s_j, a_j) \geq n$, *then the following holds*

$$\mathbb{V}[\bar{w}_{\eta}\hat{\Delta}_{SIR}^{\eta}] \leq \mathbb{V}[\bar{w}\hat{\Delta}_{SIR}]. \tag{13}$$

It indicates that $\eta < 1$ might bring a smaller variance when $\Delta(x)$ is the same for all $x \in \mathcal{D}$. However, it might not be the case and introduces a bias when $\Delta(x)$ varies across the dataset. In our experiment, we show that the performance of choosing $\hat{\Delta}_{SIR}^{1}$ is better than that of choosing $\hat{\Delta}_{SIR}^{0}$, which indicates that reducing the bias matters for resampling.

### 5.2 Other choices of IS

Other than Eqn. (4), reweighting the gradient of value function is an alternative choice to utilizing IS. The TD update of value function could be written as follows:

$$\Delta_{TD} = \mathbb{E}_{(s,a,s')\sim\mathcal{D}}[w(s,a)\nabla_{\theta}V_{\theta}(s)(V_{\theta}(s) - r(s,a) - \gamma V_{\hat{\theta}}(s'))]. \tag{14}$$

We point out that a Q-value function is still required and learned via Bellman update to learn a policy. This implementation increases computational complexity compared to IAC. In addition, learning three components simultaneously complicates the training process.

## 6 Experiments

In this section, we conduct several experiments to justify the validity of our proposed method. We aim to answer four questions: (1) Does SIR have a smaller variance than IS? (2) Does our method actually have a small extrapolation error? (3) Does our method perform better than previous methods on standard offline MuJoCo benchmarks? (4) How does each component of IAC contribute to our proposed method?

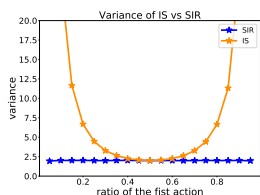

### 6.1 Variance

We first test the variance of SIR and IS on a two-arm bandit task. The first and the second action's reward distributions are $\mathcal{N}(-1, 1)$ and $\mathcal{N}(1, 1)$, respectively. We fix the dataset's size to be $100,000$. Then we vary the ratios of these two actions' samples in the dataset to simulate a set of behavior policies. For a policy that chooses the two arms with

Figure 2: SIR has a smaller variance than IS on a two-arm bandit task.

identical probability, we evaluate the policy with SIR and IS based on each behavior policy in the set. Fig.2 shows the variance of SIR and IS when provided with different behavior policies. According to Fig.2, SIR has a smaller variance than IS, especially when the dataset is highly imbalanced.

### 6.2 Extrapolation error

Table 1: Averaged normalized scores on MuJoCo locomotion on five seeds. Note that m=medium, m-r=medium-replay, r=random, m-e=medium-expert, and e=expert.

| Dataset | BC | OneStep RL | TD3+BC | CQL | IQL | IAC-w/o-$\beta$ | IAC-IS | IAC |
|---|---|---|---|---|---|---|---|---|
| halfcheetah-m-v2 | 42.0±1.7 | 50.4±0.4 | 48.3±0.3 | 47.0±0.5 | 47.4±0.2 | **52.2±0.3** | 52.0±0.6 | **51.6±0.3** |
| hopper-m-v2 | 56.2±4.3 | **87.5±10.9** | 59.3±4.2 | 53.0±28.5 | 66.2±5.7 | 83.1±23.4 | 63.8±9.8 | 74.6±11.5 |
| walker2d-m-v2 | 71.0±6.5 | **84.8±2.9** | 83.7±2.1 | 73.3±17.7 | 78.3±8.7 | 83.6±1.7 | 85.3±0.7 | 85.2±0.4 |
| halfcheetah-m-r-v2 | 36.4±2.7 | 42.7±1.3 | 44.6±0.5 | 45.5±0.7 | 44.2±1.2 | **47.5±0.5** | 47.9±0.4 | 47.2±0.3 |
| hopper-m-r-v2 | 21.8±0.5 | 98.5±2.7 | 60.9±18.8 | 88.7±12.9 | 94.7±8.6 | 102.7±1.2 | 99.3±2.7 | **103.2±1.0** |
| walker2d-m-r-v2 | 24.9±6.3 | 61.7±16.3 | 81.8±5.5 | 81.8±2.7 | 73.8±7.1 | **93.2±0.6** | 91.1±1.1 | **93.2±1.8** |
| halfcheetah-m-e-v2 | 59.6±5.8 | 75.1±14.1 | 90.7±4.3 | 75.6±25.7 | 86.7±5.3 | 89.6±2.5 | 78.1±8.2 | **92.9±0.7** |
| hopper-m-e-v2 | 51.7±2.4 | **108.6±4.1** | 98.0±9.4 | 105.6±12.9 | 91.5±14.3 | 111.0±1.6 | 107.8±5.9 | 109.3±4.0 |
| walker2d-m-e-v2 | 101.2±3.6 | 111.3±0.4 | 110.1±0.5 | 107.9±1.6 | 109.6±1.0 | **113.3±1.2** | 109.5±0.8 | 110.1±0.1 |
| halfcheetah-e-v2 | **92.9±0.5** | 88.2±6.5 | 96.7±1.1 | 96.3±1.3 | 95.0±0.5 | 94.7±0.4 | 94.4±0.3 | 94.5±0.5 |
| hopper-e-v2 | **110.9±0.3** | 106.9±4.1 | 107.8±7 | 96.5±28.0 | 109.4±0.5 | 110.7±1.7 | 111.6±0.2 | 110.6±1.9 |
| walker2d-e-v2 | 107.7±0.1 | **110.7±0.4** | 110.2±0.3 | 108.5±0.5 | 109.9±1.2 | 109.6±0.0 | 109.6±0.1 | 114.8±1.2 |
| halfcheetah-r-v2 | 2.6±0.0 | 2.3±0.0 | 11.0±1.1 | 17.5±1.5 | 13.1±1.3 | **23.2±2.3** | 21.9±1.0 | 20.9±1.2 |
| hopper-r-v2 | 4.1±0.1 | 5.6±1.6 | 8.5±0.6 | 7.9±0.4 | 7.9±0.2 | **31.5±0.3** | 27.6±9.1 | 31.3±0.3 |
| walker2d-r-v2 | 1.2±0.0 | 6.9±1.2 | 1.6±1.7 | 5.1±1.3 | 5.4±1.2 | 4.0±1.9 | 2.4±0.3 | 3.0±1.3 |
| locomotion-v2 total | 784.2 | 1041.2 | 1013.2 | 1010.2 | 1033.1 | **1149.9** | 1102.3 | **1142.4** |

In this part, we compare IAC to a baseline that replaces the resampling section with the general off-policy evaluation, which is updated by Eqn. (1). The policy update and the hyper-parameters for IAC and the baseline are the same. Although advantage-weighted regression enforces a constraint on the policy, the baseline might encounter target actions that are out of the dataset. We experiment with both methods on four tasks in D4RL (Fu et al., 2020). We plot the learned Q-values of IAC and the baseline in Fig.3. Also, we show the true Q-value of IAC by rollouting the trained policy for $1,000$ episodes and evaluating the Monte-Carlo return. The result shows that the learned Q-value of IAC is close to the true Q-value. Note that the learned Q-value is smaller than the true Q-value on walker2d-medium and

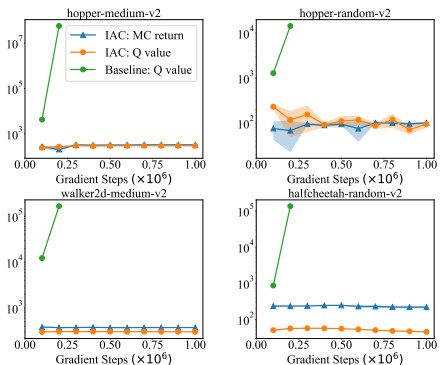

Figure 3: True Q-value of IAC, learned Q-values of IAC and a baseline without resampling.

halfcheetah-random tasks. The reason is that taking the minimum of two target networks will lead to underestimation. By contrast, the Q-value of the baseline increases fast and is far larger than that of IAC. It indicates that our proposed method has a lower extrapolation error by only referring to the target actions in the dataset.

## 6.3 COMPARISONS ON OFFLINE RL BENCHMARKS

**Gym locomotion tasks.** We evaluate our proposed approach on the D4RL benchmark (Fu et al., 2020) in comparison to prior methods (Table 1). We focus on Gym-MuJoCo locomotion domains involving three agents: halfcheetah, hopper, and walker2d. For each agent, five datasets are provided which correspond to behavior policies with different qualities: random, medium, medium-replay, medium-expert, and expert.

**AntMaze tasks.** We also compare our proposed method with prior methods in challenging AntMaze domains, which consist of sparse-reward tasks and require "stitching" fragments of suboptimal trajectories traveling undirectedly to find a path from the start to the goal of the maze. The results are shown in Table 2.

**Baselines.** Our offline RL baselines include both multi-step dynamic programming and one-step approaches. For the former, we compare to CQL (Kumar et al., 2020), TD3+BC (Fujimoto & Gu, 2021), and IQL (Kostrikov et al., 2022). For the latter, we compare to OneStep RL (Brandfonbrener et al., 2021).

Table 2: Averaged normalized scores on AntMaze on five seeds. Note that u=Umaze, u-d=Umaze-diverse, m-p=medium-replay, m-d=medium-diverse, l-p=large-replay, and l-d=large-diverse.

| Dataset | BC | OneStep RL | TD3+BC | CQL | IQL | IAC-w/o-$\beta$ | IAC-IS | IAC |
|---|---|---|---|---|---|---|---|---|
| antmaze-u-v2 | 66.8±6.7 | 54.0±3.4 | 73.0±34.0 | 82.6±5.7 | **89.6±4.2** | 71.6±10.0 | 0.0±0.0 | 77.6±3.8 |
| antmaze-u-d-v2 | 56.8±2.6 | 57.8±14.0 | 47.0±7.3 | 10.2±6.7 | 65.6±8.3 | 52.4±7.2 | 0.0±0.0 | **71.2±8.6** |
| antmaze-m-p-v2 | 0.0±0.0 | 0.0±0.0 | 0.0±0.0 | 59.0±1.6 | **76.4±2.7** | 75.0±2.7 | 33.0±15.6 | 72.0±7.6 |
| antmaze-m-d-v2 | 0.0±0.0 | 0.6±0.5 | 0.2±0.4 | 46.6±24.0 | 72.8±7.0 | 67.2±6.9 | 17.2±25.3 | **74.2±4.1** |
| antmaze-l-p-v2 | 0.0±0.0 | 0.0±0.0 | 0.0±0.0 | 16.4±17.1 | 42.0±3.8 | 42.6±3.8 | 34.0±12.8 | **57.0±7.4** |
| antmaze-l-d-v2 | 0.0±0.0 | 0.2±0.4 | 0.0±0.0 | 3.2±4.1 | **46.0±4.5** | 38.8±16.0 | 0.0±0.0 | 47.2±9.4 |
| antmaze-v2 total | 123.6 | 112.6 | 120.2 | 218.0 | 392.4 | 347.6 | 84.2 | **399.2** |

**Comparison with baselines.** On the Gym locomotion tasks, we find that IAC outperforms prior methods. On the more challenging AntMaze task, IAC performs comparably to IQL and outperforms OneStep RL by a large margin.

**Comparison with one-step method.** Note that one-step method corresponds to $\hat{\Delta}^0_{SIR}$, which samples uniformly. In the AntMaze tasks, especially the medium and large ones, few near-optimal trajectories are contained, and the reward signal is sparse. These domains require "stitching" parts of suboptimal trajectories to find a path from the start to the goal of the maze (Fu et al., 2020). Therefore, one-step approaches yield bad performance in these challenging domains where multi-step dynamic programming is essential.

We point out that using $\hat{\Delta}^1_{SIR}$ gives IAC the power of multi-step dynamic programming. At the same time, inheriting the advantages of one-step approaches, IAC uses in-sample data, thus having a low extrapolation error. As shown in Fig.4, our proposed method performs much better than choosing $\eta = 0$, which corresponds to OneStep RL.

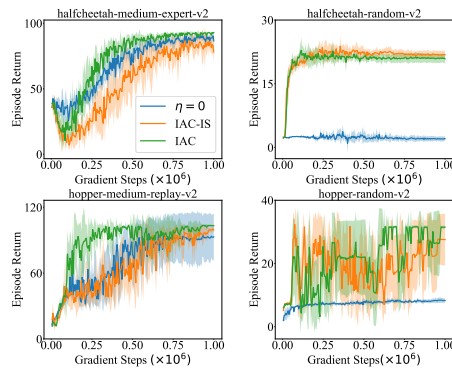

Figure 4: Comparison with OneStep RL ($\eta = 0$) and importance sampling.

**Comparison with importance sampling.** We refer to the algorithm which updates the Q-value via importance sampling(seen in Eqn. (4)) as IAC-IS, and we test its performance on MuJoCo and AntMaze tasks. The result is shown in Table 1, Table 2, and Fig.4. IAC-IS performs worse than IAC slightly on Gym locomotion tasks. For the challenging AntMaze tasks, there is a large gap between the two algorithms. IAC-IS even obtains zero rewards on half of the tasks. The reason might be that IAC-IS has a larger variance than IAC, which would impair the learning process.

**Ablation on the estimated behavior policy.** IAC requires access to a pre-trained behavior policy, which brings a computational burden. Removing the behavior policy and regarding the behavior policy density as a constant will introduce a bias but reduce the computational load. We refer to this variant as IAC-w/o-$\beta$. As shown in Table 1 and Table 2, IAC-w/o-$\beta$ could still obtain desirable performance on most Gym locomotion tasks and several AntMaze tasks. Thus, IAC-w/o-$\beta$ is an appropriate choice for its lightweight property when the computational complexity is a priority.

## 7 CONCLUSION

In this paper, we propose IAC to conduct in-sample learning by sampling-importance resampling. IAC enjoys the benefits of both multi-step dynamic programming and in-sample learning, which only relies on the target Q-values of the actions in the dataset. IAC is unbiased and has a smaller variance than importance sampling in many cases. In addition, IAC is the first method to adapt the dataset's distribution to match the trained policy dynamically during learning. The experimental results show the effectiveness of our proposed method. In future work, we expect to find a better estimated behavior policy to boost our method, such as transformers (Vaswani et al., 2017).

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

# A   SAMPLING METHODS

The problem is to find $\mu = \mathbb{E}_p f(x) = \int_{\mathcal{D}} f(x)p(x)dx$ where $p$ is the target distribution on $\mathcal{D} \subseteq \mathbb{R}^d$, when only allowed to sampling from some proposal distribution $q$ on $\mathcal{D}$. Define importance ratio $w(x) = p(x)/q(x)$

## A.1   IMPORTANCE SAMPLING

condition: $q(x) > 0$ whenever $f(x)p(x) \neq 0$, i.e., $\text{supp}(q) \supseteq \text{supp}(p \cdot f)$; $\mathbb{E}_q|w(x)f(x)| < +\infty$

$$\mathbb{E}_p(f(x)) = \int_{\text{supp}(q)} q(x)\frac{p(x)}{q(x)}f(x)dx = \mathbb{E}_q\left[f(x)w(x)\right]$$

IS estimator of $\mu$:

$$\hat{\mu}_{IS} = \frac{1}{n}\sum_{i=1}^{n} f(x_i)w(x_i), \quad x_i \sim q$$

bias:

$$\mathbb{E}_q\left(\hat{\mu}_{IS}\right) = \mu$$

variance:

$$\begin{aligned}
\text{Var}_q\left(\hat{\mu}_{IS}\right) &= \frac{\text{Var}_q(f(x)w(x))}{n} \\
&= \frac{1}{n}\int_{\mathcal{D}} \frac{(f(x)p(x))^2}{q(x)}\mathrm{d}x - \mu^2 \\
&= \frac{1}{n}\int_{\mathcal{D}} \frac{(f(x)p(x) - \mu q(x))^2}{q(x)}\mathrm{d}x
\end{aligned}$$

How to select a good proposal $p$? The numerator is small when $f(x)p(x) - \mu q(x)$ is close to zero, that is, when $q(x)$ is nearly proportional to $f(x)p(x)$. From the denominator, we see that regions with small values of $q(x)$ greatly magnify whatever lack of proportionality appears in the numerator.

**Theorem 7** (Optimality Theorem). *For fixed n, The distribution q that minimizes the variances of $\hat{\mu}_{IS}$ is*

$$q = \frac{|f(x)|p(x)}{\int |f(x)|p(x)dx} \propto |f(x)|p(x)$$

## A.2   SAMPLING-IMPORTANCE RESAMPLING

Step 1.(Sampling) Draw an independent random sample $\{x_1, \ldots, x_n\}$ from the proposal distribution $q$.

Step 2.(Importance) Calculate the importance ratio for each $x_i$: $w(x_i) = p(x_i)/q(x_i)$

Step 3.(Resampling) Draw $x^*$ from the discrete distribution over $\{x_1, \ldots, x_n\}$ with sample probabilities, $\rho(x_i) = w_i/\sum_{j=1}^{n} w_j$.

## A.3   BATCH SETTING

**Question**: If the problem is to compute $\mu = \mathbb{E}_p f(x)$, which is better, IS, SNIS, or SIR? Assume IS and SNIS have sample $B = \{x_1, \ldots, x_n\}$, while SIR resamples $k$ items batch $\tilde{b} = \{\tilde{x}_1, \ldots, \tilde{x}_k\}$ from $B$ with probability $\rho_i$ proportional to $w_i$. For fair comparison, we also consider batch version IS-b and SNIS-b, which resample $k$ items $\check{b} = \{\check{x}_1, \ldots, \check{x}_k\}$ from $B$ uniformly. The estimators are as follows,

$$\hat{\mu}_{IS} = \frac{1}{n}\sum_{i=1}^{n} f(x_i)w(x_i)$$

$$\hat{\mu}_{IS-b} = \frac{1}{k}\sum_{i=1}^{k} f(\check{x}_i)w(\check{x}_i)$$

$$\hat{\mu}_{SNIS} = \frac{\sum_{i=1}^{n} f(x_i)w(x_i)}{\sum_{i=1}^{n} w(x_i)}$$

$$\hat{\mu}_{SNIS-b} = \frac{\sum_{i=1}^{k} f(\check{x}_i)w(\check{x}_i)}{\sum_{i=1}^{k} w(\check{x}_i)}$$

$$\hat{\mu}_{SIR} = \frac{1}{k}\sum_{i=1}^{k} f(\tilde{x}_i)$$

**Proposition 8.** $\hat{\mu}_{SIR}$ *has the same bias as* $\hat{\mu}_{SNIS}$.

*Proof.*

$$\mathbb{E}_{B\sim q}\mathbb{E}_b[\hat{\mu}_{SIR}] = \mathbb{E}_{B\sim q}\mathbb{E}_b[\frac{1}{k}\sum_{j=1}^{k} f(\tilde{x}_i)] = \mathbb{E}_{B\sim q}[\mathbb{E}_b f(\tilde{x}_1)]$$

$$= \mathbb{E}_{B\sim q}\sum_{i=1}^{n} \frac{w_i}{\sum_{j=1}^{n} w_j} f(x_i)$$

$$= \mathbb{E}_{B\sim q}\hat{\mu}_{SNIS}$$

$\square$

Note that if $p$ and $q$ are normalized, $\bar{w}\hat{\mu}_{SIR}$ is unbiased, where $\bar{w} = \frac{1}{n}\sum_{i=1}^{n} w(x_i)$

**Proposition 9.** $\hat{\mu}_{SIR}$ *is consistent as* $n \to \infty$

For variance, we compare the unbiased $\bar{w}\hat{\mu}_{SIR}$ and $\hat{\mu}_{IS-b}$. On the one side, $\hat{\mu}_{IS}$ and $\hat{\mu}_{SNIS}$ use entire dataset $B$ that have much more items than batch $b$ and should have a low-variance estimate. For another, $\hat{\mu}_{SNIS-b}$ is baised and if $p$ and $q$ are normalized, the bias-corrected version of $\hat{\mu}_{SNIS-b}$ is just $\hat{\mu}_{IS-b}$.

**Proposition 10.** *For a fixed $B$, let $\mu_B = \mathbb{E}_p[f(x)|B]$. The variance of $\bar{w}\hat{\mu}_{SIR}$ and $\hat{\mu}_{IS-b}$ are as follows.*

$$\text{Var}(\hat{\mu}_{IS-b}|B) = \frac{1}{k}\left(\frac{1}{n}\sum_{j=1}^{n} w(x_j)^2\|f(x_j)\|_2^2 - \mu_B^\top\mu_B\right)$$

$$\text{Var}(\bar{w}\hat{\mu}_{SIR}|B) = \frac{1}{k}\left(\frac{\bar{w}}{n}\sum_{j=1}^{n} w_j\|f(x_j)\|_2^2 - \mu_B^\top\mu_B\right)$$

*Proof.* Since we condition on the dataset $B$, the only source of randomness is the sampling mechanism. Each index is sampled independently so we have that,

$$\text{Var}(\bar{w}\hat{\mu}_{SIR}|B) = \frac{1}{k^2}\sum_{j=1}^{k} \text{Var}\left(\bar{w}f(\tilde{x}_k)|B\right) = \frac{1}{k}\text{Var}\left(\bar{w}f(\tilde{x}_1)|B\right)$$

and similarly

$$\text{Var}(\hat{\mu}_{IS-b}|B) = \frac{1}{k}\text{Var}(w(\check{x}_1)f(\tilde{x}_1)|B)$$

We can further simplify these expressions. For the IS-b estimator

$$
\begin{aligned}
\mathrm{Var}(\hat{\mu}_{IS-b}|B) &= \frac{1}{k}\,\mathrm{Var}(w(\check{x}_1)f(\check{x}_1)|B)\\
&= \frac{1}{k}\left(\mathbb{E}[w(\check{x}_1)^2 f(\check{x}_1)^\top f(\check{x}_1)|B] - \mathbb{E}[w(\check{x}_1)f(\check{x}_1)\mid B]^\top \mathbb{E}[w(\check{x}_1)f(\check{x}_1)\mid B]\right)\\
&= \frac{1}{k}\left(\mathbb{E}[w(\check{x}_1)^2\,\|f(\check{x}_1)\|_2^2\mid B] - \mu_B^\top \mu_B\right) \quad \text{since } w(\check{x}_1)f(\check{x}_1)\mid B \text{ is unbiased for } \mu_B\\
&= \frac{1}{k}\left(\frac{1}{n}\sum_{j=1}^n w(x_j)^2\,\|f(x_j)\|_2^2 - \mu_B^\top \mu_B\right)
\end{aligned}
$$

For the SIR estimator, recalling that $\bar{w} = \frac{1}{n}\sum_{i=1}^n w_i$ , we follow similar steps,

$$
\begin{aligned}
\mathrm{Var}(\bar{w}\hat{\mu}_{SIR}|B) &= \frac{1}{k}\,\mathrm{Var}\left(\bar{w}f(\tilde{x}_1)|B\right)\\
&= \frac{1}{k}\left(\mathbb{E}[\bar{w}^2 f(\tilde{x}_1)^\top f(\tilde{x}_1)|B] - \mathbb{E}[\bar{w}f(\tilde{x}_1)\mid B]^\top \mathbb{E}[\bar{w}f(\tilde{x}_1)\mid B]\right)\\
&= \frac{1}{k}\left(\mathbb{E}[\bar{w}^2\,\|f(\tilde{x}_1)\|_2^2\mid B] - \mu_B^\top \mu_B\right) \quad \text{since } \bar{w}f(\tilde{x}_1)\mid B \text{ is unbiased for } \mu_B\\
&= \frac{1}{k}\left(\sum_{j=1}^n \bar{w}^2\,\frac{w_j}{\sum_{i=1}^n w_i}\,\|f(x_j)\|_2^2 - \mu_B^\top \mu_B\right)\\
&= \frac{1}{k}\left(\frac{\bar{w}}{n}\sum_{j=1}^n w_j\,\|f(x_j)\|_2^2 - \mu_B^\top \mu_B\right)
\end{aligned}
$$

$\square$

Under what condition SIR estimator has a lower variance than IS-b?

Condition 1: $\|f(x_j)\|_2^2 > c/w_j$ for samples where $w_j \geq \bar{w}$, and $\|f(x_j)\|_2^2 < c/w_j$ for samples where $w_j < \bar{w}$, for some $c > 0$.

## B  PROOFS

### B.1  PROOF OF PROPOSITION 1

Sampling-importance resampling (SIR) aims at drawing a random sample from a target distribution $p$. Typically, SIR consists of three steps:

Step 1.(Sampling) Draw an independent random sample $\{x_1, \ldots, x_n\}$ from the proposal distribution $q$.

Step 2.(Importance) Calculate the importance ratio for each $x_i$: $w(x_i) = p(x_i)/q(x_i)$

Step 3.(Resampling) Draw $x^*$ from the discrete distribution over $\{x_1, \ldots, x_n\}$ with sample probabilities, $\rho(x_i) = w_i/\sum_{j=1}^n w_j$.

*Proof.* $x^*$ has cdf

$$\begin{aligned}
\Pr\left(x^* \le x_0\right) &= \sum_{i=1}^n \rho_i \mathbb{I}[x_i \in (-\infty, x_0)] \\
&= \frac{\frac{1}{n}\sum_{i=1}^n w_i \mathbb{I}[x_i \in (-\infty, x_0)]}{\frac{1}{n}\sum_{i=1}^n w_i} \xrightarrow[n\to\infty]{} \frac{\mathbb{E}_q w(x)\mathbb{I}[x \in (-\infty, x_0)]}{\mathbb{E}_q w(x)} \\
&= \frac{\int_{-\infty}^{x_0} p(x)dx}{\int_{-\infty}^{\infty} p(x)dx}
\end{aligned}$$

$\square$

Note that even if $p$ and $q$ are unnormalized (but can be normalized), this method still works. The sample size under SIR can be as large as desired. The less $p$ resembles $q$, the larger the sample size $n$ will need to be in order that the distribution of $x^*$ well approximates $p$.

### B.2 PROOF OF THEOREM 3

*Proof.* Note that there are two source of randomness for the estimator $\hat{\Delta}_{SIR}$. First $\mathcal{D} = \{x_1, \ldots, x_n\}$ is sampled i.i.d. according to $p_\beta$. Second, our method draws $\tilde{x}$ i.i.d. from the discrete distribution over $\{x_1, \ldots, x_n\}$ placing mass $\rho_i$ on $x_i$, forming a mini-batch $\tilde{b} = \{\tilde{x}_1, \ldots, \tilde{x}_k\}$.

$$\begin{aligned}
\mathbb{E}_{\mathcal{D}\sim p_\beta}\mathbb{E}_{\tilde{b}\sim\rho}[\bar{w}\hat{\Delta}_{SIR}] &= \bar{w}\mathbb{E}_{\mathcal{D}\sim p_\beta}\mathbb{E}_{\tilde{b}\sim\rho}[\frac{1}{k}\sum_{j=1}^k \Delta(\tilde{x}_j)] = \bar{w}\mathbb{E}_{\mathcal{D}\sim p_\beta}[\mathbb{E}_{\tilde{x}_1\sim\rho}\Delta(\tilde{x}_1)] \\
&= \bar{w}\mathbb{E}_{\mathcal{D}\sim p_\beta}\sum_{i=1}^n \rho_i\Delta(x_i) = \bar{w}\mathbb{E}_{\mathcal{D}\sim p_\beta}\sum_{i=1}^n \frac{w_i}{\sum_{j=1}^n w_j}\Delta(x_i) \\
&= \mathbb{E}_{\mathcal{D}\sim p_\beta}\frac{1}{n}\sum_{i=1}^n w_i\Delta(x_i) = \mathbb{E}_{x\sim p_\beta}w\Delta(x) \\
&= \mathbb{E}_{x\sim p_\beta}\frac{\pi(a'|s')}{\beta(a'|s')}\Delta(x) = \mathbb{E}_{x\sim p_\beta}\frac{p_\pi(x)}{p_\beta(x)}\Delta(x) = \mathbb{E}_{x\sim p_\pi}\Delta(x) \\
&= \Delta_{TD}
\end{aligned}$$

$\square$

### B.3 PROOF OF PROPOSITION 4

*Proof.* For IS, we have $\mathbb{V}[\hat{\Delta}_{IS}] = \frac{1}{k}\left(\frac{1}{n}\sum_{j=1}^n w(x_j)^2 \|\Delta(x_j)\|_2^2 - \mu_B^\top\mu_B\right)$, where $k$ is the size of the batch and $\mu_B$ is the expectation of the estimator. Since SIR is unbiased, we have $\mathbb{V}[\bar{w}\hat{\Delta}_{SIR}] = \frac{1}{k}\left(\frac{\bar{w}}{n}\sum_{j=1}^n w_j\|\Delta(x_j)\|_2^2 - \mu_B^\top\mu_B\right)$ for SIR.

Now the problem is to show that $\sum_{j=1}^n w(x_j)^2\|\Delta(x_j)\|_2^2 \ge \bar{w}\sum_{j=1}^n w(x_j)\|\Delta(x_j)\|_2^2$.

$$\sum_{j=1}^n w(x_j)^2\|\Delta(x_j)\|_2^2 \ge \bar{w}\sum_{j=1}^n w(x_j)\|\Delta(x_j)\|_2^2 \tag{15}$$

$$\Leftrightarrow \sum_{j=1}^n \rho_j^2\|\Delta(x_j)\|_2^2 \ge \frac{1}{n}\sum_{j=1}^n \rho_j\|\Delta(x_j)\|_2^2, \tag{16}$$

where $\rho_j = \frac{w(x_j)}{\sum_{i=1}^n w(x_i)} = \frac{w(x_j)}{n\bar{w}}$.

Assume that normalized gradient is applied: $\|\Delta(x_j)\|_2^2 = 1$. Then according to Cauchy-Schwarz Inequality, Eqn. (16) hold.

$$n\sum_{j=1}^n \rho_j^2 = \left(\sum_{j=1}^n 1^2\right)\left(\sum_{j=1}^n \rho_j^2\right) \geq \left(\sum_{j=1}^n 1\times\rho_j\right)^2 = 1 = \sum_{j=1}^n \rho_j$$

$\square$

### B.4 PROOF OF PROPOSITION 5

*Proof.* Assume that $(x_1, ..., .x_n)$ is in descending order in terms of $\|\Delta(x_i)\|_2^2$, otherwise we could rearrange these items. Considering that $\|\Delta(x)\|_2^2$ is in positive relation to $\frac{1}{\beta(a'|s')}$ for $x \in \mathcal{D}$ and policy $\pi$ is uniform. To simplify notation, we use $\|f(x)\|$ to denote $\|\Delta(x)\|_2^2$. We have

$$\|f(x_1)\| \geq \|f(x_2)\| \geq \cdots \geq \|f(x_n)\|. \tag{17}$$

$$\rho_1 \geq \rho_2 \geq \cdots \geq \rho_n, (\text{or } w(x_1) \geq w(x_2) \geq \cdots \geq w(x_n)) \tag{18}$$

Let denote by $j_*$ the index of largest $\rho_j$ satisfying $\rho_j < \frac{1}{n}$, i.e.,

$$j_* = \min\left\{j : \rho_j < \frac{1}{n}, 1 \leq j \leq n\right\}$$

It is easy to show $j_* > 1$. We have $\rho_j - \frac{1}{n} \geq 0$ when $1 \leq j \leq j_* - 1$ and $\rho_j - \frac{1}{n} < 0$ when $j_* \leq j \leq n$. Then

$$\sum_{j=1}^n \rho_j \|f(x_j)\|^2 \left(\rho_j - \frac{1}{n}\right)$$

$$= \sum_{j=1}^{j_*-1} \rho_j \|f(x_j)\|^2 \left(\rho_j - \frac{1}{n}\right) + \sum_{j=j_*}^n \rho_j \|f(x_j)\|^2 \left(\rho_j - \frac{1}{n}\right)$$

$$\geq \sum_{j=1}^{j_*-1} \rho_{j_*-1} \|f(x_{j_*-1})\|^2 \left(\rho_j - \frac{1}{n}\right) + \sum_{j=j_*}^n \rho_j \|f(x_j)\|^2 \left(\rho_j - \frac{1}{n}\right)$$

$$= \rho_{j_*-1} \|f(x_{j_*-1})\|^2 \left(1 - \sum_{j=j_*}^n \rho_j - \frac{j_*-1}{n}\right) + \sum_{j=j_*}^n \rho_j \|f(x_j)\|^2 \left(\rho_j - \frac{1}{n}\right)$$

$$= \sum_{j=j_*}^n \rho_{j_*-1} \|f(x_{j_*-1})\|^2 \left(\frac{1}{n} - \rho_j\right) + \sum_{j=j_*}^n \rho_j \|f(x_j)\|^2 \left(\rho_j - \frac{1}{n}\right)$$

$$= \sum_{j=j_*}^n \left(\rho_{j_*-1} \|f(x_{j_*-1})\|^2 - \rho_j \|f(x_j)\|^2\right) \left(\frac{1}{n} - \rho_j\right) \geq 0.$$

$\square$

### B.5 PROOF OF PROPOSITION 6

Assume that $\forall x \in \mathcal{D}$, $\Delta(x) = \mathbf{h}$, where $\mathbf{h}$ is a constant vector. Let $\eta \in [0, 1]$, $\bar{w}_\eta$ denote $\frac{1}{n}\sum_{j=1}^n w(s_j, a_j)^\eta$. Assume that $\sum_{j=1}^n w(s_j, a_j) \geq n$, then the following holds

$$\mathbb{V}[\bar{w}_\eta \hat{\Delta}_{SIR}^\eta] \leq \mathbb{V}[\bar{w}\hat{\Delta}_{SIR}]. \tag{19}$$

*Proof.* we have $\mathbb{V}[\bar{w}\hat{\Delta}_{SIR}] = \frac{1}{k}\left(\frac{\bar{w}}{n}\sum_{j=1}^n w_j \|\Delta(x_j)\|_2^2\right)$,

$\mathbb{V}[\bar{w}_\eta \hat{\Delta}_{SIR}^\eta] = \frac{1}{k}\left(\frac{\bar{w}_\eta}{n}\sum_{j=1}^n w_j^\eta \|\Delta(x_j)\|_2^2\right)$.

When normalized gradient is applied, the problem is to show that

$$\bar{w}\sum_{j=1}^{n}w_j \geq \bar{w}_\eta \sum_{j=1}^{n}w_j^\eta$$

$$\Leftrightarrow \sum_{j=1}^{n}w_j \geq \sum_{j=1}^{n}w_j^\eta$$

According to Holder Inequality, we have

$$\left(\sum_{j=1}^{n}x_j^\alpha\right)^{\frac{1}{\alpha}}\left(\sum_{j=1}^{n}1^{\frac{\alpha}{\alpha-1}}\right)^{1-\frac{1}{\alpha}} \geq \sum_{j=1}^{n}x_j$$

$$\Leftrightarrow \left(\sum_{j=1}^{n}x_j^\alpha\right)^{\frac{1}{\alpha}}n^{1-\frac{1}{\alpha}} \geq \sum_{j=1}^{n}x_j$$

By choosing $x_j = w_j^\eta$ and $\alpha = \frac{1}{\eta}$, we have

$$(\sum_{j=1}^{n}w_j)^\eta n^{1-\eta} \geq \sum_{j=1}^{n}w_j^\eta$$

According to the condition $\sum_{j=1}^{n}w_j \geq n$, The following holds,

$$\sum_{j=1}^{n}w_j = (\sum_{j=1}^{n}w_j)^{\eta+1-\eta} \geq (\sum_{j=1}^{n}w_j)^\eta n^{1-\eta} \geq \sum_{j=1}^{n}w_j^\eta$$

$\square$

## C EXPERIMENTAL DETAILS AND EXTENDED RESULTS

### C.1 EXPERIMENTAL DETAILS

Table 3: Hyperparameters of policy training in IAC.

|  | Hyperparameter | Value |
| --- | --- | --- |
| IAC | Optimizer | Adam (Kingma & Ba, 2014) |
|  | Critic learning rate | $3 \times 10^{-4}$ |
|  | Actor learning rate | $3 \times 10^{-4}$ with cosine schedule |
|  | Batch size | 256 |
|  | Discount factor | 0.99 |
|  | Number of iterations | $10^6$ |
|  | Target update rate $\tau$ | 0.005 |
|  | Policy update frequency | 2 |
|  | Inverse temperature of AWR $\beta$ | {0.25, 5} for Gym-MuJoCo {10} for AntMaze |
|  | Variance of Gaussian Policy | 0.1 |
| Architecture | Actor | input-256-256-output |
|  | Critic | input-256-256-1 |

For the MuJoCo locomotion tasks, we average returns of over 10 evaluation trajectories and 5 random seeds, while for the Ant Maze tasks, we average over 100 evaluation trajectories and 5 random seeds. Following the suggestions of the authors of the dataset, we subtract 1 from the rewards for

the Ant Maze datasets. We choose TD3 (Fujimoto et al., 2018) as our base algorithm and optimize a deterministic policy. To compute the SIR/IS ratio, we need the density of any action under the deterministic policy. For this, we assume all policies are Gaussian with a fixed variance 0.1. Note that IAC has no additional hyperparameter to tune. The only hyperparameter we tuned is the inverse temperature $\beta$ in AWR for PI. We use $\beta = 10$ for Ant Maze tasks and $\beta = \{0.25, 5\}$ for MuJoCo locomotion tasks ($\beta = 0.25$ for expert and medium-expert datasets, $\beta = 5$ for medium, medium-replay, random datasets). And following previous work (Brandfonbrener et al., 2021), we clip exponentiated advantages to $(-\infty, 100]$. All hyperparameters are included in Table 3.

## C.2 Learning Curves of IAC

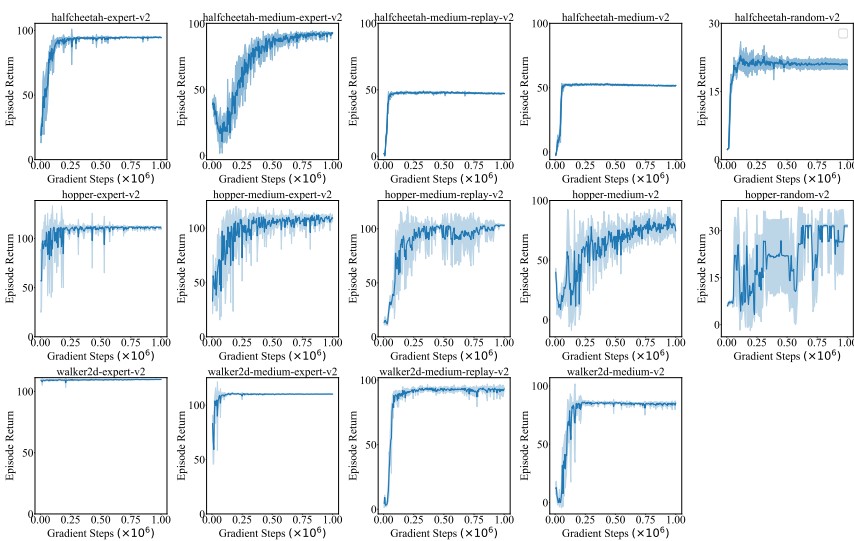

Figure 5: Learning Curves of IAC on MuJoCo Locomotion Tasks.

# D Ablation on behavior policy

## D.1 VAE behavior policy

In this section, we conduct ablation study on behavior policy. Like previous works (Fujimoto et al., 2019; Wu et al., 2022), we consider to learn the behavior density $\hat{\beta}$ explicitly using conditional variational auto-encoder (Kingma & Welling, 2013; Sohn et al., 2015). Specifically, $\beta(a|s)$ can be approximated by a Deep Latent Variable Model $p_{\omega_1}(a|s) = \int p_{\omega_1}(a|z, s) p(z|s) \mathrm{d}z$ with prior $p(z|s) = \mathcal{N}(\mathbf{0}, I)$. Rather than computing $p_{\omega_1}(a|s)$ directly by marginalization, VAE construct a lower bound on the likelihood $p_{\omega_1}(a|s)$ by introducing an approximate posterior $q_{\omega_2}(z|a, s)$:

$$
\begin{aligned}
\log p_{\omega_1}(a|s) = \log \mathbb{E}_{q_{\omega_2}(z|a,s)} \left[ \frac{p_{\omega_1}(a, z|s)}{q_{\omega_2}(z|a, s)} \right] &\geq \mathbb{E}_{q_{\omega_2}(z|a,s)} \left[ \log \frac{p_{\omega_1}(a, z|s)}{q_{\omega_2}(z|a, s)} \right] \\
&= \mathbb{E}_{q_{\omega_2}(z|a,s)} \left[ \log p_{\omega_1}(a|z, s) \right] - \mathrm{KL} \left[ q_{\omega_2}(z|a, s) \| p(z|s) \right] \\
&\stackrel{\text{def}}{=} J_{\text{ELBO}}(s, a; \omega).
\end{aligned}
\tag{20}
$$

It converts the difficult computation problem into an optimization problem. Instead of maximizing the log-likelihood $\log p_{\omega_1}(a|s)$ directly, we now optimize parameters $\omega_1$ and $\omega_2$ jointly by maximizing the evidence lower bound (ELBO) $J_{\text{ELBO}}(s, a; \omega)$. After pre-training the VAE, we simply use $J_{\text{ELBO}}(s, a; \omega)$ to approximate $\log \beta(a|s)$ in Eqn. (7).

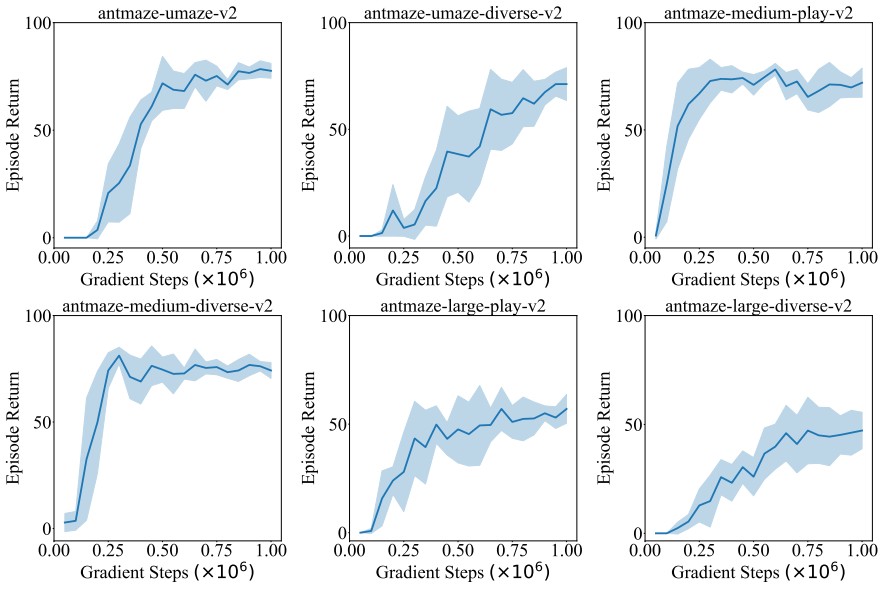

Figure 6: Learning Curves of IAC on AntMaze Tasks.

Table 4: Averaged normalized scores on MuJoCo locomotion on five seeds. We compare IAC with IAC-MM(IAC with multi modal behavior policy), IAC-SR(IAC with state ratio), IAC-VAE(IAC with VAE behavior policy), and IAC-SNIS(IAC with self normalized importance sampling). Note that m=medium, m-r=medium-replay, r=random, m-e=medium-expert, and e=expert.

| Dataset | IAC-MM | IAC-SR | IAC-VAE | IAC-SNIS | IAC |
|---|---|---|---|---|---|
| halfcheetah-m-v2 | 51.9±0.0 | 51.4±0.3 | 52.2±0.2 | 51.3±0.3 | 51.6±0.3 |
| hopper-m-v2 | 88.9±12.7 | 61.5±0.3 | 91.9±18.3 | 82.9±10.2 | 74.6±11.5 |
| walker2d-m-v2 | 84.8±0.7 | 84.7±1.0 | 85.2±0.5 | 82.8±0.7 | 85.2±0.4 |
| halfcheetah-m-r-v2 | 47.5±0.0 | 46.4±0.0 | 47.6±0.4 | 47.9±0.6 | 47.2±0.3 |
| hopper-m-r-v2 | 103.9±0.2 | 100.9±1.1 | 103.1±1.1 | 99.9±1.0 | 103.2±1.0 |
| walker2d-m-r-v2 | 90.9±1.0 | 92.7±1.6 | 93.6±0.4 | 92.7±0.3 | 93.2±1.8 |
| halfcheetah-m-e-v2 | 92.7±0.4 | 94.7±1.6 | 87.8±7.4 | 87.1±9.2 | 92.9±0.7 |
| hopper-m-e-v2 | 108.3±0.0 | 109.4±0.5 | 111.6±0.5 | 107.8±0.7 | 109.3±4.0 |
| walker2d-m-e-v2 | 110.2±0.1 | 110.1±0.1 | 109.9±0.5 | 109.7±0.4 | 110.1±0.1 |
| halfcheetah-e-v2 | 94.6±0.0 | 94.5±0.0 | 94.6±0.3 | 94.9±0.2 | 94.5±0.5 |
| hopper-e-v2 | 111.0±0.5 | 111.1±0.1 | 111.4±0.4 | 111.1±0.0 | 110.6±1.9 |
| walker2d-e-v2 | 109.8±0.1 | 109.4±0.1 | 109.6±0.1 | 109.7±0.2 | 114.8±1.2 |
| halfcheetah-r-v2 | 21.3±0.6 | 19.2±0.8 | 21.0±0.7 | 23±0.9 | 20.9±1.2 |
| hopper-r-v2 | 20.4±11.1 | 19.2±9.5 | 31.3±0.0 | 32±0.2 | 31.3±0.3 |
| walker2d-r-v2 | 0.4±0.2 | 7.5±7.7 | 1.3±2.3 | 0±0.0 | 3.0±1.3 |
| locomotion-v2 total | 1135.8 | 1112.7 | 1152.1 | 1132.8 | 1142.4 |

We term this variant IAC-VAE. The results of IAC-VAE are shown in Table 4 and Table 5. Benefiting from the VAE estimator, IAC-VAE obtains better results in MuJoCo locomotion tasks.

## D.2 CATEGORICAL BEHAVIOR POLICY

Modeling the behavior policy as a unimodal Gaussian distribution will limit its flexibility and representation ability. We consider capturing multi modes of the behavior policy. To that end, We

Table 5: Averaged normalized scores on AntMaze on five seeds. We compare IAC with IAC-MM, IAC-SR, IAC-VAE, and IAC-SNIS. Note that u=Umaze, u-d=Umaze-diverse, m-p=medium-replay, m-d=medium-diverse, l-p=large-replay, and l-d=large-diverse.

| Dataset | IAC-MM | IAC-SR | IAC-VAE | IAC-SNIS | IAC |
|---|---|---|---|---|---|
| antmaze-u-v2 | 52.7±2.9 | 72.1±5.0 | 84.0±6.6 | 40.0±9.2 | 77.6±3.8 |
| antmaze-u-d-v2 | 35.7±10.2 | 65.3±6.9 | 55.0±6.1 | 2.0±3.5 | 71.2±8.6 |
| antmaze-m-p-v2 | 47.2±4.1 | 70.1±9.4 | 71.0±4.4 | 78.0±7.5 | 72.0±7.6 |
| antmaze-m-d-v2 | 1.6±0.3 | 65.2±7.2 | 71.7±0.6 | 72.0±7.8 | 74.2±4.1 |
| antmaze-l-p-v2 | 15.9±2.0 | 51.2±5.9 | 41.0±11.5 | 47.3±10.0 | 57.0±7.4 |
| antmaze-l-d-v2 | 6.6±1.3 | 40.1±11.3 | 51.3±6.5 | 30.0±10.5 | 47.2±9.4 |
| antmaze-v2 total | 159.8 | 364 | 374 | 269.3 | 399.2 |

experiment by modeling the action space as a discrete. Considering that the action range is [-1, 1], we split each action into 40 categories and each category has a range of 0.05. Then the behavior policy is estimated by cross-entropy. In this setting, the behavior policy is multi-modal.

We term this variant IAC-MM. We compare this variant and IAC and the result is shown in Table 4 and Table 5. The result shows in several settings the variant has marginally better performance. But the overall performance is worse compared to IAC. Especially in AntMaze tasks, IAC-MM suffers a performance drop. The reason might be that the classification task ignores the relation between nearby actions.

## E DISCRETE DOMAIN

We also test IAC and IAC-w/o-$\beta$ on the CartPole task which has discrete action space. The dataset contains the samples in the replay buffer when we train a discrete SAC(soft-actor-critic) until convergence. The result is shown in Fig. 7. IAC-w/o-$\beta$ has a much worse final performance than IAC. Also, the maximum performance during training is worse than IAC.

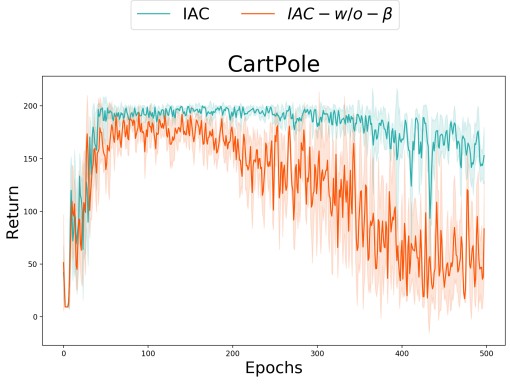

Figure 7: Learning Curves of CartPole task.

## F OTHER BENCHMARKS

To make a comprehensive comparison, we also compare AWAC and CRR with IAC. The results of MuJoCo locomation and AntMaze tasks are shown in Table 6 and Table 7, respectively. The results show that our methods have better performance than these baselines.

Table 6: Averaged normalized scores on MuJoCo locomotion on five seeds. Other than the baselines above, we compare with AWAC and CRR. Note that m=medium, m-r=medium-replay, r=random, m-e=medium-expert, and e=expert.

| Dataset | BC | OneStep RL | TD3+BC | CQL | IQL | AWAC | CRR | IAC |
|---|---|---|---|---|---|---|---|---|
| halfcheetah-m-v2 | 42.0±1.7 | 50.4±0.4 | 48.3±0.3 | 47.0±0.5 | 47.4±0.2 | 47.9±0.1 | 47.1±0.1 | 51.6±0.3 |
| hopper-m-v2 | 56.2±4.3 | 87.5±10.9 | 59.3±4.2 | 53.0±28.5 | 66.2±5.7 | 59.8±0.7 | 38.1±1.8 | 74.6±11.5 |
| walker2d-m-v2 | 71.0±6.5 | 84.8±2.9 | 83.7±2.1 | 73.3±17.7 | 78.3±8.7 | 83.1±1.6 | 59.7±0.7 | 85.2±0.4 |
| halfcheetah-m-r-v2 | 36.4±2.7 | 42.7±1.3 | 44.6±0.5 | 45.5±0.7 | 44.2±1.2 | 44.8±0.1 | 44.4±0.3 | 47.2±0.3 |
| hopper-m-r-v2 | 21.8±0.5 | 98.5±2.7 | 60.9±18.8 | 88.7±12.9 | 94.7±8.6 | 69.8±0.1 | 25.5±1.6 | 103.2±1.0 |
| walker2d-m-r-v2 | 24.9±6.3 | 61.7±16.3 | 81.8±5.5 | 81.8±2.7 | 73.8±7.1 | 78.1±5.6 | 27.0±0.7 | 93.2±1.8 |
| halfcheetah-m-e-v2 | 59.6±5.8 | 75.1±14.1 | 90.7±4.3 | 75.6±25.7 | 86.7±5.3 | 64.9±2.3 | 85.2±1.9 | 92.9±0.7 |
| hopper-m-e-v2 | 51.7±2.4 | 108.6±4.1 | 98.0±9.4 | 105.6±12.9 | 91.5±14.3 | 100.1±9.9 | 53.0±5.1 | 109.3±4.0 |
| walker2d-m-e-v2 | 101.2±3.6 | 111.3±0.4 | 110.1±0.5 | 107.9±1.6 | 109.6±1.0 | 110.0±0.2 | 91.3±11.4 | 110.1±0.1 |
| halfcheetah-e-v2 | 92.9±0.5 | 88.2±6.5 | 96.7±1.1 | 96.3±1.3 | 95.0±0.5 | 81.7±4.2 | 93.5±0.7 | 94.5±0.5 |
| hopper-e-v2 | 110.9±0.3 | 106.9±4.1 | 107.8±7 | 96.5±28.0 | 109.4±0.5 | 109.5±1.5 | 108.7±3.0 | 110.6±1.9 |
| walker2d-e-v2 | 107.7±0.1 | 110.7±0.4 | 110.2±0.3 | 108.5±0.5 | 109.9±1.2 | 110.1±0.0 | 108.9±0.5 | 114.8±1.2 |
| halfcheetah-r-v2 | 2.6±0.0 | 2.3±0.0 | 11.0±1.1 | 17.5±1.5 | 13.1±1.3 | 6.1±0.2 | 13.6±1.1 | 20.9±1.2 |
| hopper-r-v2 | 4.1±0.1 | 5.6±1.6 | 8.5±0.6 | 7.9±0.4 | 7.9±0.2 | 9.2±0.6 | 16.1±6.0 | 31.3±0.3 |
| walker2d-r-v2 | 1.2±0.0 | 6.9±1.2 | 1.6±1.7 | 5.1±1.3 | 5.4±1.2 | 0.2±0.7 | 4.9±0.8 | 3.0±1.3 |
| locomotion-v2 total | 784.2 | 1041.2 | 1013.2 | 1010.2 | 1033.1 | 975.6 | 817 | 1142.4 |

Table 7: Averaged normalized scores on AntMaze on five seeds. Other than the baselines above, we compare with AWAC and CRR. Note that u=Umaze, u-d=Umaze-diverse, m-p=medium-replay, m-d=medium-diverse, l-p=large-replay, and l-d=large-diverse.

| Dataset | BC | OneStep RL | TD3+BC | CQL | IQL | AWAC | CRR | IAC |
|---|---|---|---|---|---|---|---|---|
| antmaze-u-v2 | 66.8±6.7 | 54.0±3.4 | 73.0±34.0 | 82.6±5.7 | 89.6±4.2 | 80.0±1.7 | 43.8±2.3 | 77.6±3.8 |
| antmaze-u-d-v2 | 56.8±2.6 | 57.8±14.0 | 47.0±7.3 | 10.2±6.7 | 65.6±8.3 | 52.0±6.9 | 42.8±1.2 | 71.2±8.6 |
| antmaze-m-p-v2 | 0.0±0.0 | 0.0±0.0 | 0.0±0.0 | 59.0±1.6 | 76.4±2.7 | 0.0±0.0 | 0.4±0.0 | 72.0±7.6 |
| antmaze-m-d-v2 | 0.0±0.0 | 0.6±0.5 | 0.2±0.4 | 46.6±24.0 | 72.8±7.0 | 0.2±0.2 | 0.5±0.2 | 74.2±4.1 |
| antmaze-l-p-v2 | 0.0±0.0 | 0.0±0.0 | 0.0±0.0 | 16.4±17.1 | 42.0±3.8 | 0.0±0.0 | 0.0±0.0 | 57.0±7.4 |
| antmaze-l-d-v2 | 0.0±0.0 | 0.2±0.4 | 0.0±0.0 | 3.2±4.1 | 46.0±4.5 | 0.0±0.0 | 0.0±0.0 | 47.2±9.4 |
| antmaze-v2 total | 123.6 | 112.6 | 120.2 | 218 | 392.4 | 132.2 | 87.6 | 399.2 |

# G    OTHER ABLATIONS

## G.1    ABLATION ON STATE RATIO

Using state-distribution correction $\frac{d^\pi(s)}{d^\beta(s)}$ might be helpful to IAC. For most RL settings, the dimension of the state is larger than that of the action. Since high dimensional estimation is challenging, it is difficult to estimate $d^\pi(s)$ and $d^\beta(s)$. Thus we sort to estimate $\frac{d^\pi(s)}{d^\beta(s)}$ by following the paper 'Infinite-Horizon Off-Policy Estimation'. We term this variant IAC-SR. The result is shown in Table 4 and Table 5. It indicates that introducing state-distribution correction worsens the performance. One reason is that the approximation for $\frac{d^\pi(s)}{d^\beta(s)}$ is not accurate and it will introduce bias to the algorithm.

## G.2    ABLATION ON SELF-NORMALIZED IMPORTANCE SAMPLING

Considering that self-normalized importance sampling also has a lower variance than importance sampling, we test the performance of a variant with self-normalized importance sampling. This variant is termed IAC-SNIS. The result of IAC-SNIS is shown in Table 4 and Table 5. The self-normalized importance sampling variant performs comparably to IAC on MuJoCo tasks but performs worse than IAC on AntMaze tasks.

## G.3    ABLATION ON $\eta$

To study the hyperparameter $\eta$'s effect on our proposed method, We run the experiments of using different $\eta$ in $\{0.1, 0.3, 0.5, 0.7, 0.9\}$. The result is shown in Fig. 8. It can be seen that the variant with a large $\eta$ performs better than with a small $\eta$ .

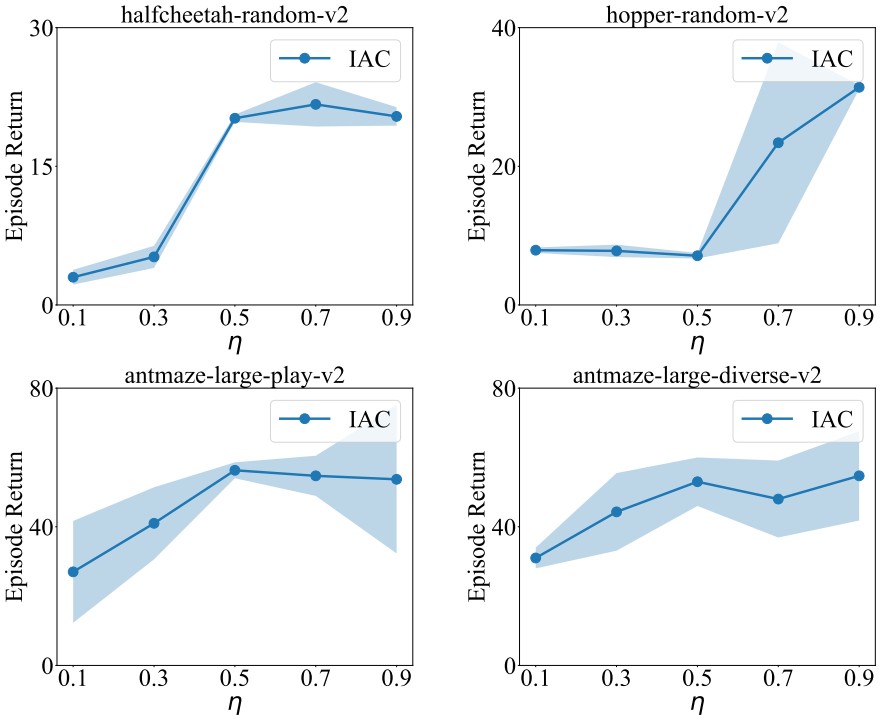

Figure 8: The effect of $\eta$ on IAC.

Table 8: Runtime of TD3BC, CQL, IQL, IAC for halfcheetah-medium-replay on a GeForce RTX 3090.

| Algorithm | TD3BC | IQL | CQL | IAC | pre-training in IAC |
|-----------|-------|-----|-----|-----|---------------------|
| Runtime | 1h | 1h50min | 4h10min | 2h30min | 2min |

### G.4 RUNTIME

We test the runtime of IAC on halfcheetah-medium-replay on a GeForce RTX 3090. The results of IAC and other baselines are shown in Table 8. It takes 2h30min for IAC to finish the task, which is comparable to other baselines. Note that it only takes two minutes for the pre-training part.

