# OpenReview forum: "In-sample Actor Critic for Offline Reinforcement Learning"
_ICLR.cc/2023/Conference — ICLR 2023 poster_

### Official Review · Reviewer_nmzD · 2022-10-23

**Confidence:** 4
**Correctness:** 4
**Technical Novelty And Significance:** 3
**Empirical Novelty And Significance:** 2
**Recommendation:** 8

**Clarity, Quality, Novelty And Reproducibility:**

- Quality: high. The proposed method is well-presented. It is theoretically analyzed in a sound way, and the empirical evaluation is thorough and well-designed (with strong baselines and challenging environments)
- Clarity: medium-high. The paper is easy to follow for someone very familiar with offline RL algorithms, but I have the feeling that a quick reader, or someone less familiar with offline RL, may find the paper less clear. For instance, neither the abstract nor the introduction mention the simple summary of the paper: "we modify how samples are sampled from the experience buffer to correct for out-of-distribution actions". Sentences like "utilizes sampling-importance resampling to execute in-sample policy evaluation" are only understandable by people already familiar with sampling-importance resampling.
- Originality: average. The proposed method seems novel, and sampling-importance resampling is not used often, but it is quite close to Prioritized ER (mentioned by the authors in the paper). This does not lower the value of the paper, though.

**Strength And Weaknesses:**

Strengths:

- The problem being considered is important and an active area of research. The proposed solution is elegant and outperforms the baselines. It also seems relatively easy to implement and computationally cheap (except that a SumTree needs to be added to the experience buffer)
- The paper is clear and well-written, and theoretical details are given to motivate the approach, and justify the results.
- Section 4.4, the practical implementation, is very well-written and concisely gives all the details needed for implementing the algorithm

Weaknesses:

- I did not find any big weakness of this work, only minor comments.
- Starting from Equation 4, the notation mixes $\Delta$ and $\nabla$, without a (one-line) explanation of what these symbols mean. It is a bit weird to compute a TD-error from a gradient, so the equation looks like the $\Delta$ is also a gradient (the gradient of the Q update, not the update itself). I find this quite confusing and explanations of the notations would be welcome.

Question:

- For offline RL papers, I'm always curious of what happens if the proposed algorithm is used in an online setting. In the case of this paper, is it possible to, every so often, let $\pi_{\theta}$ execute an episode in the environment, add the resulting transitions to the experience buffer, and continue learning from that?

**Summary Of The Paper:**

The paper considers the offline Reinforcement Learning setting, and more precisely the problem of computing Q-Values from transitions in the dataset, where bootstrapping from out-of-distribution actions (actions not in the dataset) may lead to over-estimation and poor learning. The proposed solution builds on sampling importance-resampling, and changes how the transitions used for learning are sampled from the dataset, to sample "good" (in-distribution) samples more often. These transitions are used to train a critic (using a somewhat conventional TD loss, with the main contribution being the sampling and not a new loss), and an actor, in the loss of which the estimated behavior policy of the dataset appears.

Experimental results are thorough and show that the proposed method outperforms several baselines on challenging continuous-action tasks.

**Summary Of The Review:**

The paper considers an important problem, proposes an elegant solution with a theoretical analysis, and empirically shows that the proposed method outperforms many baselines.

---

> ### Author Response · Authors · 2022-11-18
> **Response to Reviewer nmzD**
>
> Thanks for your meaningful comment.
>
> ### Q.1 The notation mixes $\Delta$ and $\nabla$.
>
> Sorry for the unclear statement. As you point out, $\Delta_{TD}$ is the gradient of the TD-error (Eq (1)). Vanilla policy evaluation uses this to update the Q network paramters $\theta$. In this paper we try to estimate $\Delta_{TD}$ with completely in-dataset $a'$ to avoid extrapolation error. To emphasize that, we use the symbol $\Delta_{TD}$ to denote the actual gradient of the TD-error (Eq (1)) and use symbol $\\Delta_{SIR}$ to denote our estimate of $\Delta_{TD}$. $\nabla$ is the gradient operator and $\nabla_\theta Q_\theta(s,a)$ denotes the gradient of the Q-value function.
>
>
>
> ### Q.2 What happens if the proposed algorithm is used in an online setting. In the case of this paper, is it possible to, every so often, let $\pi_\theta$ execute an episode in the environment, add the resulting transitions to the experience buffer, and continue learning from that?
>
> Thanks for pointing it out. Since the distribution of online samples differs from offline samples, continuing learning from an offline agent will also suffer from distribution shift. Specially, when only learning from the online samples, the performance will have a drop at first due to distribution shift and then increase during training. When learning by uniformly sampling from offline samples and online samples, the large offline dataset will slow down the progress. A strategy is to  sample online samples uniformly and sample offline samples according their on-policy levels[1]. To be specific, for offline dataset, one can sample according to the probability $\frac{d_{on}(s,a)}{d_{\beta}(s,a)} $, where $d_{on}(s,a)$ is the distribution of online replay buffer and $d_{\beta}(s,a)$ is the distribution of offline replay buffer.
>
> [1] Lee, Seunghyun, et al. "Offline-to-online reinforcement learning via balanced replay and pessimistic q-ensemble." Conference on Robot Learning. PMLR, 2022.

---

### Official Review · Reviewer_uBTY · 2022-10-27

**Confidence:** 4
**Correctness:** 4
**Technical Novelty And Significance:** 2
**Empirical Novelty And Significance:** 3
**Recommendation:** 5

**Clarity, Quality, Novelty And Reproducibility:**

The paper is generally well-written and easy to follow. It seems that the technical novelty is limited given that IAC is a combination of existing methods for policy evaluation and policy improvement. It seems Theorem 3 of the paper is very similar to Corollary 3.1.1 in [1], but it was not properly mentioned.

[1] Schlegel et al., Importance Resampling for Off-policy Prediction, 2019


**Strength And Weaknesses:**

[Strengths]
1. This paper presents an offline RL algorithm exploiting in-sample learning based on importance resampling, which enables stable learning yet achieves good empirical performance.


[Weaknesses]
1. It seems technical novelty is limited, given that IAC can be understood of a combination of Importance Resampling [1] for policy evaluation and AWAC[2]/CRR[3] for the policy improvement part.
2. The algorithm requires behavior density estimation, which introduces additional computational costs and hyperparameters.
3. In the experiments, comparison with some baselines is missing, e.g. AWAC and CRR[3], which are using similar approaches. Also, it would be great to compare with the method using self-normalized importance sampling instead of SIR.


[Questions and comments]
1. I am wondering if using state-distribution correction $d^\pi(s) / d^\mu(s)$ can further improve the performance.
2. In addition to the OneStepRL and IQL, OptiDICE [4] only requires in-distribution samples $(s,a,r,s')$ of the offline dataset too.
3. I am curious to see the result when using different $\eta$ in Eq (12), e.g. $\{0.1, 0.3, 0.5, 0.7, 0.9\}$. Does $\eta=1$ always perform the best?


[1] Schlegel et al., Importance Resampling for Off-policy Prediction, 2019

[2] Peng et al., Advantage-weighted regression: Simple and scalable off-policy reinforcement learning, 2019

[3] Weng et al., Critic Regularized Regression, 2020

[4] Lee et al., OptiDICE: Offline Policy Optimization via Stationary Distribution Correction Estimation, 2021


**Summary Of The Paper:**

This paper presents In-sample Actor-Critic (IAC), an algorithm for offline RL. The policy evaluation of IAC is done only using in-distribution samples of the dataset, where each sample's (normalized) importance ratio is used for resampling probabilities. Then, TD updates are performed using the resampled samples. For policy improvement, advantage-weighted regression is used to ensure that the learning policy samples actions within dataset support. The SIR is a consistent estimator and has a lower variance compared to the standard importance sampling. Experimental results show that IAC performs competitively with the state-of-the-art methods on Gym-Mujoco locomotion tasks and AntMaze tasks.


**Summary Of The Review:**

I think this paper presents a sound offline RL algorithm of in-sample learning. My main concerns are technical novelty and some missing baselines in the experiments.

---

> ### Author Response · Authors · 2022-11-18
> **Response to Reviewer uBTY**
>
> Thanks for your constructive comment.
>
> ### Q.1  IAC can be understood of a combination of Importance Resampling for policy evaluation and AWAC/CRR for the policy improvement part.
>
> Sorry for the unclear statement. The contribution of IAC lies in that it conducts unbiased in-sample learning for offline RL compared to previous works. The Importance Resampling [1] aims to estimate a value function from a behavior policy and does not aim for offline reinforcement learning. Our key observation is that for offline reinforcement learning, the out-of-sample target action for the Q function will bring extrapolation error. We adopt importance resampling to get rid of out-of-sample action and avoid extrapolation error. From the implementation side, IAC resamples according to $\frac{\pi(a'|s')}{\beta(a'|s')}$ rather than $\frac{\pi(a|s)}{\beta(a|s)}$ in [1]. It is actually aimed at offline RL to avoid out-of-sample target action.
>
>
> ### Q.2 The algorithm requires behavior density estimation, which introduces additional computational costs and hyperparameters.
>
> Thanks for pointing it out. We test the runtime of IAC on halfcheetah-medium-replay on a GeForce RTX 3090. The results of IAC and other baselines are shown in Table 8 in the revised version. It takes 2h30min for IAC to finish the task, which is comparable to other baselines. Note that it only takes two minutes for the pre-training part. Baselines such as BCQ and BEAR also have the behavior policy training part, which use more complicated VAE architecture. Also, as we point out, IAC-w/o-$\beta$ can produce comparable results to  IAC.
>
> ### Q.3 In the experiments, comparison with some baselines is missing, e.g. AWAC and CRR[3]. Also, it would be great to compare with the method using self-normalized importance sampling instead of SIR.
>
> Thanks a lot for your suggestion. We supplement the result of AWAC and CRR in Table 6 and Table 7 in the revised version. As the result shows, IAC has better performance than AWAC and CRR on MuJoCo and AntMaze tasks. Considering that self-normalized importance sampling also has a lower variance than importance sampling, we test the performance of a variant with self-normalized importance sampling. This variant is termed IAC-SNIS. The result of IAC-SNIS is shown in Table 4 and Table 5 in the revised version. The self-normalized importance sampling variant performs comparably to IAC on MuJoCo tasks but performs worse than IAC on AntMaze tasks.
>
> ### Q.4 If using state-distribution correction $\frac{d\pi(s)}{d\mu(s)}$ can further improve the performance.
>
> Thanks for your proposal. IAC is unbiased and does not need the  state-distribution correction term, which is the difference between IAC and other off-policy evaluation works.
>
> We test the influence of state-distribution correction $\frac{d\pi(s)}{d\mu(s)}$ on IAC. For most RL settings, the dimension of the state is larger than that of the action. Since high dimensional estimation is challenging, it is difficult to estimate $d\pi(s)$ and $d\mu(s)$. Thus we sort to estimate $\frac{d\pi(s)}{d\mu(s)}$ by following [2] We term this variant IAC-SR. The result is shown in Table 4 and Table 5 in the revised version. It indicates that introducing state-distribution correction worsens the performance. One reason is that the approximation for $\frac{d\pi(s)}{d\mu(s)}$) is not accurate and it will introduce bias to the algorithm.
>
> ### Q.5 In addition to the OneStepRL and IQL, OptiDICE only requires in-distribution samples $(s,a,r,s')$ of the offline dataset too.
>
> We regret missing the relevant paper. Thanks a lot for providing the valuable reference.
>
> ### Q.6 What is the result when using different $\eta$ in Eq (12), e.g. 0.1,0.3,0.5,0.7,0.9. Does $\eta$=1 always perform the best?
>
> Thanks for your proposal. We run the experiments using different $\eta$ in {0.1,0.3,0.5,0.7,0.9}. The results are shown in Figure 8 in the revised version. It can be seen that the variant with a large $\eta$ performs better than that with a small $\eta$.
>
> [1] Schlegel et al., Importance Resampling for Off-policy Prediction, 2019
>
> [2] Liu, Qiang, et al. "Breaking the curse of horizon: Infinite-horizon off-policy estimation." Advances in Neural Information Processing Systems 31 (2018).

---

### Official Review · Reviewer_Y1J6 · 2022-10-27

**Confidence:** 4
**Correctness:** 3
**Technical Novelty And Significance:** 3
**Empirical Novelty And Significance:** 2
**Recommendation:** 6

**Clarity, Quality, Novelty And Reproducibility:**



The presentation of the paper is clear.

The methods in this paper fit well into the line of in-sample offline RL learning.

The implementation of the paper is not open-sourced. Open-source code is not compulsory, however, it could help readers understand some details of the methods.

**Details Of Ethics Concerns:**



**Strength And Weaknesses:**



Strength:

The method is very clear and makes sense for in-sample offline learning.

The paper contains several solid theoretical analyses of the method.

The experiments on Gym locomotion tasks and ANtMaze tasks are sufficient to show the effectiveness of the method.

Weakness:

One limitation of the method is that it needs to know the probability of the in-sample action in the dataset. Although the paper also presented the result by removing the density estimator, this method does not perform well on AntMaze problem. Can you give the reason why the method does not perform well on the AntMaze-UMaze and AntMaze-Large datasets? The paper also didn't report the training time for using a density estimator.

The density estimator only uses Gaussian distribution to model the distribution. This often could limit the capacity as in practice the distribution of the dataset could hardly be a unimodal distribution.



Several Questions:

What's the training time for using a density estimator?

I cannot see why the proposed method outperforms other in-sample offline RL methods. Could the author provide some insights on this?



**Summary Of The Paper:**

This paper applied the sampling-importance resampling methods to offline learning to conduct in-sample learning. The methods in this paper fit well into the line of in-sample offline RL learning.



**Summary Of The Review:**



The presentation of the paper is very clear and makes sense for in-sample offline learning. The theoretical analyses of the method are solid and well support the main idea of the paper. The experiments on various tasks are also sufficient to prove the effectiveness of the method.

Although the method proposed in this paper is very interesting and contributes to the community, I cannot see why the superiority of the proposed method compared to existing other in-sample methods.

---

> ### Author Response · Authors · 2022-11-18
> **Response to Reviewer Y1J6**
>
> Thanks for your valuable comment.
>
> ### Q.1 One limitation of the method is that it needs to know the probability of the in-sample action in the dataset. Although the paper also presented the result by removing the density estimator, this method does not perform well on AntMaze problem. Can you give the reason why the method does not perform well on the AntMaze-UMaze and AntMaze-Large datasets.
>
> Thanks for pointing it out. The behavior policy’s density does not vary wildly for MuJoCo tasks. So IAC-w/o-$\beta$ obtains a similar result to IAC. For Antmaze tasks which have a similar number of samples and have higher dimensions of state and action spaces, the behavior policy’s density varies rapidly. Treating $\beta$ as a constant will introduce a bias and hurt the performance.
>
> ### Q.2 The density estimator only uses Gaussian distribution to model the distribution. This often could limit the capacity as in practice the distribution of the dataset could hardly be a unimodal distribution.
>
> Thanks for your comment. We experiment by modeling the action space as a discrete space. Considering that the action range is [-1, 1], we split each action into 40 categories and each category has a range of 0.05. Then the behavior policy is estimated by cross-entropy. In this setting, the behavior policy is multi-modal. We term this variant IAC-MM. We compare this variant and IAC and the result is shown in Table 4 and Table 5 in the revised version. The result shows in several settings the variant has marginally better performance. But the overall performance is worse compared to IAC.
>
>
> ### Q.3 What's the training time for using a density estimator?
>
> Thanks for pointing it out. We test the runtime of IAC on halfcheetah-medium-replay on a GeForce RTX 3090. The results of IAC and other baselines are shown in Table 8 in the revised version. It takes 2h30min for IAC to finish the task, which is comparable to other baselines. Note that it only takes two minutes for the pre-training part. Baselines such as BCQ and BEAR also have the behavior policy training part, which use a more complicated VAE architecture. Also, as we point out, IAC-w/o-$\beta$ can produce comparable results to  IAC.
>
>
> ### Q.4 Why dose the proposed method outperform other in-sample offline RL methods?
>
> Sorry for the unclear explanation. Other in-sample offline methods contain One-step RL and IQL. One-step RL does not conduct dynamic programming and only improves the policy a step over the Q-value function of the behavior policy, and thus has worse performance. IQL’s disadvantage is that it only calculates unbiased in-sample maximum when the expectile is 1. However, when the expectile is 1, IQL suffers from instability. Thus a smaller expectile such as 0.9 is adopted by IQL and a biased solution is obtained. Compared to these methods, IAC performs unbiased in-sample learning and has better performance.

---

### Official Review · Reviewer_Ga1X · 2022-10-29

**Confidence:** 5
**Clarity, Quality, Novelty And Reproducibility:** Please see the previous section for m…
**Correctness:** 3
**Technical Novelty And Significance:** 3
**Empirical Novelty And Significance:** 3
**Recommendation:** 5

**Strength And Weaknesses:**

Strength

As discussed in the paper, offline RL algorithms suffer from extrapolation error by bootstrapping from out-of-distribution (OOD) actions. Previous methods mostly rely on using regularization to penalize OOD actions. But such methods are very sensitive to the regularization level. More importantly, when the data collection policy is very bad, the regularization actually prevents the algorithm from finding a good policy. What I really like about this paper is to rethink the problem from the first principle: is it possible to do bootstrapping from the in-sample actions? From this perspective I think this paper makes contribution to a very important problem in offline RL. The paper is also well organized and well written. Due to the high variance of importance sampling (IS), the paper proposes to use sampling-importance resampling (RIS). The authors then compare the variance of RIS and IS, and discuss how RIS is applied to the critic learning objective in offline RL algorithms. Most implementation details are clearly discussed in the paper.


================================================


Weakness:

I have the following questions and comments. Please correct me if I missed anything important. I am happy to adjust my score based on how well the authors answer the questions in the rebuttal.

1. My major concern is on the computation cost of the algorithm. It seems that after updating the current policy, IAC needs to adjust the weights (is ratio) for all (s,a) in the dataset.

2. IAC needs to learn the behavior policy before training. How does it affect the performance if we train $\beta$ as the other networks? Is there an ablation study for this?

3. The paper suggests that one can simply treat $\beta$ as a constant such that it’s canceled in the weights. In fact, from Table 1, we can see that w/o $\beta$ even produce better results than original IAC. Is there an explanation on this? It’s always interesting to understand why something you try to approximate actually works better than the original algorithm.

4. Following the previous question, I am wondering if it is because the domains are continuous control problems, where we can only observe one action in one state. Thus the log probability of $\beta$ will be very close to zero. Then I am wondering how does IAC perform in discrete problems?

5. IAC works very well empirically. But I am wondering why the algorithm performs so well, especially compared to IQL, which also performs in-sample bootstrapping. One limits of IQL in my opinion is that it’s hard to approximate the in-sample maximum with expectile regression when the behavior policy is very sub-optimal. But it seems to me that RIS also has the problem. I think it’s better to provide some evidence, even simple proof-of-concept experiments, to show the advantage of IAC over IQL.

6. In Section 5.1, the authors suggest that IAC outperforms one-step RL as it has lower bias. But isn’t that the main advantage of IAC compared to one-step RL is to use multi-step DP?  (This doesn’t effect my score, I am just wondering if this discussion is important. )

7. Finally, I am wondering what does OOD action exactly mean? The paper seems to suggest that OOD actions are those actions that are not in the dataset. Despite this is very reasonable, another view is to think of actions that are out of the support of the data collection policy. The difference is that for example an action can be covered by the behavior policy, but due to finite samples we don’t observe it in the dataset. It seems to be the second perspective is more interesting, as it allows generalization. Any comments on this?


**Summary Of The Paper:**

This paper proposes In-sample Actor Critic (IAC), a new algorithm for offline reinforcement learning. The main idea is to avoid extrapolation error by explicitly performing in-sample policy evaluation to learn the critic. To implement this, IAC applies sampling-importance resampling which has lower variance than a naive importance sampling implementation. The paper provides empirical studies in Mujoco locomotion control and AntMaze showing that IAC contains competitive performance compared to the state-of-the-art algorithms.

**Summary Of The Review:**

I recommend not accepting as the proposed algorithm might have large computational cost in practice and some claims are not well support by the experiment results. However, I am willing to adjust my score if the authors answer my questions in the rebuttal.

---

> ### Author Response · Authors · 2022-11-18
> **Response to Reviewer Ga1X**
>
> Thanks for your meaningful comment.
>
> ### Q.1 Computation cost of the algorithm.
>
> Sorry that we miss the details. Similar to PER(prioritized experience replay) which adjusts the priority of a mini-batch during training, we adjust the importance ratio of the mini-batch, which reduces the computation cost. We test the runtime of IAC on halfcheetah-medium-replay on a GeForce RTX 3090. The results of IAC and other baselines are shown in Table 8 in the revised version. It takes 2h30min for IAC to finish the task, which is comparable to other baselines. We also test the performance of adjusting the weights of a larger portion of a dataset at each iteration and finds small differences. The reason might be that for deep models, a small learning rate(3e-4) will change the policy mildly during training.
>
>
> ### Q.2 IAC needs to learn the behavior policy before training. How does it affect the performance if we train $\beta$ as the other networks? Is there an ablation study for this?
>
> Thanks for your proposal. We compare with a variant that models the behavior policy as a variational autoencoder. VAE behavior policy is widely used in offline RL, such as BCQ and BEAR. We refer to this variant as IAC-VAE. The implementation details are described in Section D.1 and the results are shown in Table 4, and Table 5 in the revised version. The variant of VAE behavior policy obtains comparable performance to IAC.
>
> In addition, the behavior policy might be multi-modal. We model the action space as a discrete space. Considering that the action range is [-1, 1], we split each action into 40 categories and each category has a range of 0.05. Then the behavior policy is estimated by cross-entropy. In this setting, the behavior policy is multi-modal. We term this variant IAC-MM. The result is shown in Table 4 and Table 5 in the revised version. The result shows in several settings the variant has marginally better performance. But the overall performance is worse compared to IAC.
>
>
> ### Q.3 The paper suggests that one can simply treat $\beta$ as a constant such that it’s canceled in the weights. In fact, from Table 1, we can see that w/o $\beta$ even produce better results than original IAC. Is there an explanation on this?
>
> Thanks a lot for pointing out the issue. We have tried to explore this issue when we observe the phenomenon. One reason is that density estimation is still challenging in the machine learning field. In addition, the behavior policy’s density does not vary wildly for MuJoCo tasks. So the variant produces a similar result to IAC. For Antmaze tasks which have a similar number of samples and have higher dimension of state and action spaces, the behavior policy’s density varies rapidly. Treating $\beta$ as a constant will introduce a bias and hurt the performance.
>
>
> ### Q.4 Following the previous question, if it is because the domains are continuous control problems, where we can only observe one action in one state. Thus the log probability of $\beta$ will be very close to zero. How does IAC perform in discrete problems?
>
> Thanks for your suggestion. As the former question, the reason might be the challenging density estimation. We also test IAC and IAC-w/o-$\beta$  on the CartPole task which has discrete action space. The dataset contains the samples in the replay buffer when we train a discrete SAC(soft-actor-critic) until convergence. The result is shown in Figure 7 in the revised version. IAC-w/o-$\beta$ has a much worse final performance than IAC. Also, the maximum performance during training is worse than IAC.
>
>
> ### Q.5 Why IAC performs so well, especially compared to IQL.
>
> We would like to just make an emphasis on that even the behavior policy is not very sub-optimal, IQL still  cannot capture the actual in-sample maximum. The reason is that to guarantee stability, IQL must select a $\tau$ which is smaller than 1 and leads to biased optimization. Our proposed IAC performs unbiased in-sample learning.
>
>
> ### Q.6 In Section 5.1, the authors suggest that IAC outperforms one-step RL as it has lower bias. But isn’t that the main advantage of IAC compared to one-step RL is to use multi-step DP?
>
> We agree with your comment that the main advantage of IAC compared to one-step RL is to use multi-step DP. Section 5.1 just discusses the influence of $\eta$ on our estimator.
>
>
> ### Q.7 What does OOD action exactly mean?
>
> Sorry for the unclear statement. OOD actions and out-of-sample actions are different. OOD actions are those out of support of the data collection policy. Out-of-sample actions are actions not in the dataset. OOD actions belong to out-of-sample actions. Note that although in-distribution actions allow more generalization, in-distribution actions will bring extrapolation error as long as they are out-of-sample. Other methods rely on different kinds of constraints to mitigate this issue (BEAR, CQL). IAC is based on the idea of getting rid of extrapolation error completely.

---

### Decision · Program_Chairs · 2023-01-20

**Decision:**

Accept: poster

**Justification For Why Not Higher Score:**

The limited technical novelty and experiments are restricted to only continuous control domains.

**Justification For Why Not Lower Score:**

The paper is very well-written, the method is well-justified with some interesting theoretical and empirical results. The authors did a great job at addressing the concerns raised by the reviewers during the rebuttal. This paper could be an interesting contribution to the ICLR community interested in offline RL.

**Metareview: Summary, Strengths And Weaknesses:**

## Summary

Offline RL methods often suffers from OOD state-action pairs and extrapolation errors. Most ORL approaches address this issue by penalizing the OOD state-action pairs and regularize policy to stay close to the behavior policy. This paper proposes an in-sample actor critic (IAC) method to perform in-sample policy evaluation via importance resampling which samples in-distribution transitions more often. The proposed method has lower variance than vanilla importance sampling based approaches. The paper provides empirical results on continuous control tasks like locomotion  and AntMaze domains.

Below I will list some of the strengths and weaknesses as indicated by the reviewers:

## Strengths
- The paper is well-written and easy to follow.
- Attacking an important problem in offline RL.
- The results seem to be strong on the tasks that the IAC was evaluated on.
- Interesting theoretical results and well-supported justification for the method.

## Weaknesses
- Limited technical novelty in terms of proposed algorithm or approach.
- More thorough analysis of the computational cost in the paper is needed.
- Results are only limited to continuous control problems. It is not clear how they would generalize to the discrete actions.
- Some unclear statements such as what is meant by the OOD actions.
- The algorithm relies on a behavior density estimation, which introduces additional computational costs and hyperparameters.

*Decision:* Offline RL is an important problem and this paper is attacking and important issue in offline RL with respect to extrapolation error and OOD actions. The experimental results are encouraging and the paper is very well-written. The authors did a great job addressing the concerns raised by reviewers. Thus I am recommending the paper for acceptance with the following recommended changes:

1. Please add a discussion about the training time and computational complexity of the approach. How much it differs in terms of FLOPS or GPU-hours for instance when compared against a One-Step RL approach. This was also raised by two of the reviewers.

2. Please clarify the unclear statements raised by the reviewers.

3. In the related work please cite [1] which came before (Brandfonbrener et al., 2021) and proposed one-step RL methods for domains with discrete actions. (Brandfonbrener et al., 2021) work extended it to the continuous control domains as they mentioned in their paper. Also [2]   does an in-sample offline actor-critic approach via a hybrid approach by training the value function as a BVE to estimate the behavior value but still uses v-trace to correct the off-policiness in the policy updates. Please discuss [2] in the related works as well.

### References

[1]: Gulcehre, C., Colmenarejo, S.G., Wang, Z., Sygnowski, J., Paine, T., Zolna, K., Chen, Y., Hoffman, M., Pascanu, R. and de Freitas, N., 2021. Regularized behavior value estimation. arXiv preprint arXiv:2103.09575.

[2]: Mathieu, M., Ozair, S., Srinivasan, S., Gulcehre, C., Zhang, S., Jiang, R., Le Paine, T., Zolna, K., Powell, R., Schrittwieser, J. and Choi, D., 2021, October. StarCraft II Unplugged: Large Scale Offline Reinforcement Learning. In Deep RL Workshop NeurIPS 2021.

**Note From Pc:**

if the above contains the word "oral" or "spotlight" please see: "oral" presentation means -> notable-top-5% and "spotlight" means -> notable-top-25%. As stated in our emails, we are disassociating presentation type from AC recommendations